# A Unified Generalization Analysis of Re-Weighting and Logit-Adjustment for Imbalanced Learning

**Zitai Wang**[1,2]     **Qianqian Xu**[3*]     **Zhiyong Yang**[4]
**Yuan He**[5]     **Xiaochun Cao**[6]     **Qingming Huang**[4,3,7*]

[1] SKLOIS, Institute of Information Engineering, CAS
[2] School of Cyber Security, University of Chinese Academy of Sciences
[3] Key Lab. of Intelligent Information Processing, Institute of Computing Tech., CAS
[4] School of Computer Science and Tech., University of Chinese Academy of Sciences
[5] Alibaba Group
[6] School of Cyber Science and Tech., Shenzhen Campus of Sun Yat-sen University
[7] BDKM, University of Chinese Academy of Sciences
wangzitai@iie.ac.cn     xuqianqian@ict.ac.cn
yangzhiyong21@ucas.ac.cn     heyuan.hy@alibaba-inc.com
caoxiaochun@mail.sysu.edu.cn     qmhuang@ucas.ac.cn

## Abstract

Real-world datasets are typically imbalanced in the sense that only a few classes have numerous samples, while many classes are associated with only a few samples. As a result, a naïve ERM learning process will be biased towards the majority classes, making it difficult to generalize to the minority classes. To address this issue, one simple but effective approach is to modify the loss function to emphasize the learning on minority classes, such as re-weighting the losses or adjusting the logits via class-dependent terms. However, existing generalization analysis of such losses is still coarse-grained and fragmented, failing to explain some empirical results. To bridge this gap, we propose a novel technique named data-dependent contraction to capture how these modified losses handle different classes. On top of this technique, a fine-grained generalization bound is established for imbalanced learning, which helps reveal the mystery of re-weighting and logit-adjustment in a unified manner. Furthermore, a principled learning algorithm is developed based on the theoretical insights. Finally, the empirical results on benchmark datasets not only validate the theoretical results but also demonstrate the effectiveness of the proposed method.

## 1 Introduction

In recent years, machine learning has achieved great success with the help of well-collected datasets, where the number of samples is artificially balanced among classes [1, 2]. However, the real-world datasets are generally imbalanced in the sense that only a few classes have numerous samples (*i.e.*, the majority ones), while the others are associated with only a few samples (*i.e.*, the minority ones) [3–5]. Owing to this issue, a naïve Empirical Risk Minimization (ERM) learning process will be biased towards the majority classes, and the generalization on the minority ones becomes challenging. Hence, the imbalanced learning problem has attracted increasing attention in recent years [6–9].

One simple yet effective approach for imbalanced learning is to modify the naïve loss function, such that the learning process can pay more attention to the minority classes (Please refer to Appendix A for more orthogonal approaches). In this direction, existing approaches generally fall into two categories: re-weighting [10, 11] and logit-adjustment [12–16]. The former category assigns larger weights to

---

*Corresponding authors.

37th Conference on Neural Information Processing Systems (NeurIPS 2023).

the losses of the minority classes. Although intuitive, this approach might lead to difficulties and instability in optimization [11, 12, 17]. To tackle this issue, Cao et al. [12] propose an effective scheme named Deferred Re-Weighting (DRW), where the re-weighting approach is applied only during the terminal phase of training. The latter category adjusts the logits by class-dependent terms. For example, the Label Distribution Aware Margin (LDAM) loss enforces larger margins for minority classes to achieve strong regularization [12]. The Logit-Adjustment (LA) loss [13] and the Class-Dependent Temperatures (CDT) loss [14] utilize additive and multiplicative terms to adjust the logits, respectively. Most recently, Kini et al. [16] combine the two types of terms and proposes a unified loss named Vector-Scaling (VS) for imbalanced learning.

Although existing loss-modification methods have achieved promising performance, the theoretical insights are still fragmented and coarse-grained. To be specific, Cao et al. [12] and Ren et al. [18] utilize the classic margin theory to explain the necessity of the additive terms in the LDAM loss. However, the theory fails to explain the significant improvement induced by the DRW scheme. Menon et al. [13] analyzes the Fisher consistency property [19] of the additive terms in the LA loss, while providing no further generalization analysis. Kini et al. [16] provides a generalization analysis of the VS loss, but the results can only explain the role of the multiplicative terms under the assumption that a linear model is trained on linearly separable data. Besides, we find that the VS loss is rather incompatible with the DRW scheme, which is also out of the scope of existing theory. Hence, a gap still exists between the theory and the practice of the loss-modification approaches.

To bridge this gap, this paper provides a systematical and fine-grained analysis of loss-modification approaches. After revisiting prior arts, we find that the only property of the loss function utilized in existing proofs is the classic Lipschitz continuity [19, 20]. However, this property is global in nature such that the whole analysis provides no insight into how the losses handle different classes. Inspired by this observation, we extend the classic Lipschitz continuity with a local technique. In this way, the local Lipschitz constants on different classes exactly correspond to the class-dependent terms of the modified loss functions. And a fine-grained generalization bound is established by a novel technique named data-dependent contraction. By applying this bound to the VS loss, the mystery of re-weighting and logit-adjustment is finally uncovered. Last but not least, a principled learning algorithm is proposed based on our theoretical insights.

To sum up, the main contributions of this paper are listed as follows:

- **New technique.** We extend the classic Lipschitz continuity and propose a novel technique named data-dependent contraction to obtain a fine-grained generalization bound for imbalanced learning.
- **Theoretical insights.** Based on the fine-grained bound, a systematical analysis succeeds in explaining the role of re-weighting and logit-adjustment in a unified manner, as well as some empirical results that are out of the scope of existing theories.
- **Principled Algorithm.** A principled algorithm is proposed based on the insights, where the re-weighting term is aligned with the generalization bound, and the multiplicative logit-adjustment term is removed during the DRW phase to avoid the incompatibility between terms.
- **Empirical Validation.** The empirical results on multiple benchmark datasets not only validate the theoretical results, but also demonstrate the superiority of the proposed method.

## 2 Preliminary

We first introduce the basic notations and the imbalanced learning problem in Sec.2.1. Then, we briefly review existing generalization analysis for imbalanced learning in Sec.2.2.

### 2.1 Notations and Problem Definition

We assume that the samples are drawn *i.i.d.* from a product space $\mathcal{Z} = \mathcal{X} \times \mathcal{Y}$, where $\mathcal{X}$ is the input space and $\mathcal{Y} = \{1, \cdots, C\}$ is the label space. Let $\mathcal{S} = \{(\boldsymbol{x}^{(n)}, y^{(n)})\}_{n=1}^{N}$ be the imbalanced training set sampled from the imbalanced distribution $\mathcal{D}$ defined on $\mathcal{Z}$, $\mathcal{S}_y = \{\boldsymbol{x} \mid (\boldsymbol{x}, y) \in \mathcal{S}\}$ be the set of samples from the class $y$, $N_y := |\mathcal{S}_y|$ denote the size of $\mathcal{S}_y$, and $\pi_y := N_y/N$. Without loss of generality, we assume that $N_1 \geq N_2 \geq \cdots \geq N_C$.

Let $\mathcal{D}_{\mathrm{bal}}$ be the balanced distribution defined on $\mathcal{Z}$. Specifically, a class $y$ is first uniformly sampled from $\mathcal{Y}$, and then the input $\boldsymbol{x}$ is sampled from the class-conditional distribution $\mathcal{D}_y := \mathbb{P}[\boldsymbol{x} \mid y]$.

Then, our task is to learn a score function $f : \mathcal{X} \to \mathbb{R}^C$ to minimize the risk defined on the balanced distribution:

$$\mathcal{R}_{\text{bal}}(f) := \frac{1}{C} \sum_{y=1}^{C} \mathcal{R}_y(f) = \frac{1}{C} \sum_{y=1}^{C} \mathbb{E}_{\boldsymbol{x} \sim \mathcal{D}_y} \left[ M(f(\boldsymbol{x}), y) \right], \tag{1}$$

where $\mathcal{R}_y$ is the risk defined on the class $y$, and $M : \mathbb{R}^C \times \mathcal{Y} \to \mathbb{R}_+$ is the measure that evaluates the model performance at $\boldsymbol{z} \in \mathcal{Z}$. For example, one of the most popular choices is to check whether the top-1 prediction is right: $M(f(\boldsymbol{x}), y) = \mathbf{1} \left[ y \notin \arg\max_{y' \in \mathcal{Y}} f(\boldsymbol{x})_{y'} \right]$, where $\mathbf{1}[\cdot]$ is the indicator function. Since $M$ is generally non-differential and thus hard to optimize, one has to select a differential surrogate loss $L : \mathbb{R}^C \times \mathcal{Y} \to \mathbb{R}_+$, which induces the following surrogate risk:

$$\widehat{\mathcal{R}}^L_{\text{bal}}(f) := \frac{1}{C} \sum_{y=1}^{C} \mathcal{R}^L_y(f) = \frac{1}{C} \sum_{y=1}^{C} \mathbb{E}_{\boldsymbol{x} \sim \mathcal{D}_y} \left[ L(f(\boldsymbol{x}), y) \right]. \tag{2}$$

Let $\mathcal{G} := \{ L \circ f : f \in \mathcal{F} \}$ denote the hypothesis set. Next, we consider a family of loss functions named Vector-Scaling (VS) [16]:

$$L_{\text{VS}}(f(\boldsymbol{x}), y) = -\alpha_y \log \left( \frac{e^{\beta_y f(\boldsymbol{x})_y + \Delta_y}}{\sum_{y'} e^{\beta_{y'} f(\boldsymbol{x})_{y'} + \Delta_{y'}}} \right). \tag{3}$$

The advantage behind this loss family is two-fold. On one hand, the VS loss generalizes popular re-weighting and logit-adjustment methods. For example, when $\alpha_y = 1, \beta_y = 1, \Delta_y = 0$, it becomes the traditional CE loss [19]. When $\beta_y = 1, \Delta_y = 0$, re-weighting terms $\alpha_y = \pi_y^{-1}$ and $\alpha_y = (1-p)/(1-p^{N_y}), p \in (0,1)$ recover the classic balanced loss [10] and Class-Balanced (CB) loss [11], respectively. $\alpha_y = 1, \beta_y = 1, \Delta_y = \tau \log \pi_y, \tau > 0$ yield the LA loss [13]. When $\alpha_1 = 1, \beta_y = (N_y/N_1)^\gamma, \Delta_y = 0, \gamma > 0$, we can deduce the CDT loss [14]. On the other hand, an ideal surrogate loss should be Fisher consistent such that minimizing $\mathcal{R}^L_{\text{bal}}(f)$ not only can put more emphasis on minority classes, but also helps bound $\mathcal{R}_{bal}(f)$ [19, 21]. Fortunately, prior arts [13] have shown that a subset of the VS loss family satisfies such a property:

$$L_{\text{Fisher}}(f(\boldsymbol{x}), y) := \frac{\delta_y}{\pi_y} \log[1 + \sum_{y' \neq y} \frac{\delta_{y'}}{\delta_y} e^{f(\boldsymbol{x})_{y'} - f(\boldsymbol{x})_y}], \tag{4}$$

where $\delta_y$ is an arbitrary positive constant.

## 2.2 Existing Generalization Analysis for Imbalanced Learning

In balanced learning, we can directly minimize the empirical balanced risk defined on the balanced datasets $\mathcal{S}_{\text{bal}}$ sampled from $\mathcal{D}_{\text{bal}}$:

$$\widehat{\mathcal{R}}^L_{\text{bal}}(f) := \frac{1}{N} \sum_{(\boldsymbol{x}, y) \in \mathcal{S}_{\text{bal}}} L(f(\boldsymbol{x}), y). \tag{5}$$

Then, the generalization guarantee is available by traditional concentration techniques [19]. However, in imbalanced learning, we can only minimize the empirical risk on the imbalanced dataset $\mathcal{S}$:

$$\widehat{\mathcal{R}}^L(f) := \frac{1}{N} \sum_{(\boldsymbol{x}, y) \in \mathcal{S}} L(f(\boldsymbol{x}), y). \tag{6}$$

To handle this issue, Cao et al. [12] and Ren et al. [18] aggregate the class-wise generalization bound directly with a union bound over class-wise results [19]:

**Proposition 1** (Union bound for Imbalanced Learning [12])**.** *Given the function set $\mathcal{F}$ and a loss function $L : \mathbb{R}^C \times \mathcal{Y} \to [0, M]$, then for any $\delta \in (0, 1)$, with probability at least $1 - \delta$ over the training set $\mathcal{S}$, the following generalization bound holds for all $g \in \mathcal{G}$:*

$$\mathcal{R}^L_{bal}(f) = \frac{1}{C} \sum_{y=1}^{C} \mathcal{R}^L_y(f) \precsim \frac{1}{C} \sum_{y=1}^{C} \left( \hat{\mathcal{R}}^L_y(f) + \hat{\mathfrak{C}}_{\mathcal{S}_y}(\mathcal{G}) + 3M \sqrt{\frac{\log 2C/\delta}{2N_y}} \right), \tag{7}$$

*where $\hat{\mathcal{R}}^L_y(f)$ is the empirical risk on $\mathcal{S}_y$; $\hat{\mathfrak{C}}_{\mathcal{S}}(\mathcal{G}) := \mathbb{E}_{\boldsymbol{\xi}}[\sup_{g \in \mathcal{G}} \frac{1}{N} \sum_{n=1}^{N} \xi^{(n)} g(\boldsymbol{z}^{(n)})]$ denotes the empirical complexity of the function set $\mathcal{G}$, and $\boldsymbol{\xi} := (\xi^{(1)}, \xi^{(2)}, \cdots, \xi^{(N)})$ are sampled from independent distributions such as the uniform distribution with $\{1, -1\}$; $\precsim$ denotes the asymptotic notation that omits undominated terms, that is, $f(t) \precsim g(t) \Longleftrightarrow \exists$ a constant $C > 0, f(t) \leq C \cdot g(t)$.*

To further bound the complexity term $\hat{\mathfrak{C}}_{\mathcal{S}_y}(\mathcal{G})$, Cao et al. [12] assume that the loss function $L$ satisfies the Lipschitz continuity and applies the traditional contraction lemma [20]:

**Definition 1** (Lipschitz Continuity). *Let $\| \cdot \|$ denote the 2-norm. Then, we say the loss function $L(f, y)$ is Lipschitz continuous with constant $\mu$ if for any $f, f' \in \mathcal{F}$, $\boldsymbol{x} \in \mathcal{S}$,*

$$|L(f, y) - L(f', y)| \leq \mu \cdot \|f(\boldsymbol{x}) - f'(\boldsymbol{x})\|. \tag{8}$$

**Lemma 1** (Contraction Lemma). *Assume that the loss function $L(f, \boldsymbol{x})$ is Lipschitz continuous with a constant $\mu$. Then, the following inequality holds:*

$$\hat{\mathfrak{C}}_{\mathcal{S}}(\mathcal{G}) \leq \mu \cdot \hat{\mathfrak{C}}_{\mathcal{S}}(\mathcal{F}). \tag{9}$$

Finally, the standard margin-based generalization bound [22] is directly applied to obtain the upper bound of $\hat{\mathfrak{C}}_{\mathcal{S}_y}(\mathcal{F})$. However, this union bound has the following limitations:

- Theoretically, this generalization bound is coarse-grained and not sharp enough. To be specific, the differences among different loss functions lie in the choice of $\alpha_y, \beta_y, \Delta_y$. However, the Lipschitz continuity, which is the only property of $L$ utilized in the proof, is global in nature and thus obscures these differences. Although the margin theory can provide some theoretical insights into the role of $\Delta_y$, the roles of $\alpha_y, \beta_y$ are still a mystery. Besides, since

$$\texttt{Bound}(\mathcal{R}_{\text{bal}}^L(f)) = \frac{1}{C}\texttt{Bound}(\sum_y \mathcal{R}_y^L(f)) \leq \frac{1}{C}\sum_y \texttt{Bound}(\mathcal{R}_y^L(f)), \tag{10}$$

  a sharper bound might be available if we can bound $\mathcal{R}_{\text{bal}}^L(f)$ directly.

- Empirically, although the induced LDAM loss outperforms the CE loss, the improvement is not so significant. Fortunately, when combining the Deferred Re-Weighting (DRW) technique [12], where $\alpha_y = (1 - p)/(1 - p^{N_y}), p \in (0, 1)$ [11] during the terminal phase of training, the improvement becomes much more impressive. However, Eq.(7) fails to explain this phenomenon.

Recently, Kini et al. [16] provide a generalization analysis for the VS loss. However, the results, which only hold for linear models with linearly separable data, can only explain the roles of $\beta_y$. For the role of $\Delta_y$, they resort to analyzing the gradient of the VS loss and provide a coarse-grained analysis.

To sum up, existing generalization analysis for imbalanced learning is coarse-grained and fragmented. Next, we aim to build a more fine-grained and systematical generalization bound that can unify the roles of both re-weighting and logit-adjustment.

## 3   Fine-Grained Generalization Analysis for Imbalanced Learning

In Sec.3.1, we first establish a sharp generalization bound based on a novel technique named data-dependent contraction. Then, in Sec.3.2, we apply this generalization bound to the VS loss to provide a series of theoretical insights. Finally, in Sec.3.3, a principled algorithm is proposed based on the theoretical insights.

### 3.1   Generalization Bound Induced By Data-Dependent Contraction

Different from Eq.(7), we hope to build a direct bound between $\mathcal{R}_{\text{bal}}^L(f)$ and $\widehat{\mathcal{R}}^L(f)$. To this end, our analysis is based on the following lemma, whose proof can be found in Appendix B:

**Lemma 2.** *Given the function set $\mathcal{F}$ and a loss function $L : \mathbb{R}^C \times \mathcal{Y} \to [0, M]$, then for any $\delta \in (0, 1)$, with probability at least $1 - \delta$ over the training set $\mathcal{S}$, the following generalization bound holds for all $g \in \mathcal{G}$:*

$$\mathcal{R}_{bal}^L(f) \precsim \Phi(L, \delta) + \frac{1}{C\pi_C} \cdot \hat{\mathfrak{C}}_{\mathcal{S}}(\mathcal{G}), \tag{11}$$

*where $\Phi(L, \delta) := \frac{1}{C\pi_C}[\widehat{\mathcal{R}}^L(f) + 3M\sqrt{\frac{\log 2/\delta}{2N}}]$ contains the empirical risk on $\mathcal{S}$ and the $\delta$ term.*

**Remark 1.** *Recall that $\pi_C := N_C/N, N_1 \geq N_2 \geq \cdots \geq N_C$. Hence, this lemma reveals how the model performance depends on the imbalance degree of the data.*

As shown in Sec.2.2, the fine-grained analysis is unavailable due to the global nature of the classic Lipschitz continuous property. In view of this, we extend this traditional definition with a local technique [23]:

**Definition 2** (Local Lipschitz Continuity). *We say the loss function $L(f, y)$ is local Lipschitz continuous with constants $\{\mu_y\}_{y=1}^C$ if for any $f, f' \in \mathcal{F}, y \in \mathcal{Y}, \boldsymbol{x} \in \mathcal{S}_y,$*

$$|L(f, y) - L(f', y)| \leq \mu_y \cdot \|f(\boldsymbol{x}) - f'(\boldsymbol{x})\|. \tag{12}$$

Then, the following data-dependent contraction inequality helps us obtain a sharper bound, whose proof is given in Appendix C.

**Assumption 1.** *Next, we assume that $\hat{\mathfrak{C}}_\mathcal{S}(\mathcal{F}) \sim \mathcal{O}(1/\sqrt{N})$. Note that this result holds for kernel-based models with traditional techniques [19] and neural networks with the latest techniques [24, 25]. And the prior arts also adopt this assumption [12].*

**Lemma 3** (Data-Dependent Contraction). *Assume that the loss function $L(f, \boldsymbol{x})$ is local Lipschitz continuous with constants $\{\mu_y\}_{y=1}^C$. Then, the following inequality holds under Asm.1:*

$$\hat{\mathfrak{C}}_\mathcal{S}(\mathcal{G}) \precsim \hat{\mathfrak{C}}_\mathcal{S}(\mathcal{F}) \sum_{y=1}^C \mu_y \sqrt{\pi_y}, \tag{13}$$

Combining Lem.2 and Lem.3, we have the following theorem:

**Theorem 1** (Data-Dependent Bound for Imbalanced Learning). *Given the function set $\mathcal{F}$ and a loss function $L : \mathbb{R}^C \times \mathcal{Y} \to [0, M]$, for any $\delta \in (0, 1)$, with probability at least $1 - \delta$ over the training set $\mathcal{S}$, the following generalization bound holds for all $f \in \mathcal{F}$:*

$$\mathcal{R}_{bal}^L(f) \precsim \Phi(L, \delta) + \frac{\hat{\mathfrak{C}}_\mathcal{S}(\mathcal{F})}{C\pi_C} \sum_{y=1}^C \mu_y \sqrt{\pi_y}. \tag{14}$$

At the first glance, Eq.(14) seems a little loose since $\sum_{y=1}^C \sqrt{\pi_y} > 1$. In fact, this intuition holds when local Lipschitz continuity degenerates to Def.1. However, when $\mu_y$ is decreasing *w.r.t.* $\pi_y$, a shaper bound might be available. To build an intuitive understanding, we present the following proposition, whose proof can be found in Appendix D.

**Proposition 2.** *Assume that $\mu_y \propto N_y^{-\kappa}, \kappa > 0$. Then, when $\kappa > 1$, the data-dependent bound presented in Thm.1 is sharper than the union bound defined in Prop.1.*

### 3.2 Application to the VS Loss

Next, we apply Thm.1 to the VS loss to reveal the role of both re-weighting and logit-adjustment. To this end, it is necessary to analyze the local Lipschitz property of the VS loss, whose proof is presented in Appendix E.

**Lemma 4.** *Assume that the score function is bounded. Then, the VS loss is local Lipschitz continuous with constants $\{\mu_y\}_{y=1}^C$, where*

$$\mu_y = \alpha_y \tilde{\beta}_y \left[1 - softmax\left(\beta_y B_y(f) + \Delta_y\right)\right], \tag{15}$$

$\tilde{\beta}_y := \sqrt{\beta_y^2 + \left(\sum_{y' \neq y} \beta_{y'}\right)^2}$; *softmax $(\cdot)$ denotes the softmax function; $B_y(f)$ denotes the minimal prediction on the ground-truth class $y$, i.e., $B_y(f) := \min_{\boldsymbol{x} \in S_y} f(\boldsymbol{x})_y$.*

**Remark 2.** *$B_y(f)$ is closely related to the minimal margin defined by $margin_y^\downarrow := \min_{\boldsymbol{x} \in S_y}(f(\boldsymbol{x})_y - \max_{j \neq y} f(\boldsymbol{x})_j)$. It is not difficult to check that $B_y(f) - margin_y^\downarrow \leq \max_{\boldsymbol{x} \in S_y, j \neq y} f(\boldsymbol{x})_j$. Hence, as we improve the model performance on class $y$, the RHS of the above inequality, i.e., the gap between $B_y(f)$ and $margin_y^\downarrow$ will decrease, and both the minimal margin and $B_y(f)$ will increase.*

Then, combining Thm.1 and Lem.4, we have the following proposition, which reveals how the existing loss-oriented methods improve generalization performance by exploiting the data priors.

**Proposition 3** (Data-Dependent Bound for the VS Loss). *Given the function set $\mathcal{F}$ and the VS loss $L_{VS}$, for any $\delta \in (0,1)$, with probability at least $1 - \delta$ over the training set $\mathcal{S}$, the following generalization bound holds for all $f \in \mathcal{F}$:*

$$\mathcal{R}_{bal}^L(f) \precsim \Phi(L_{VS}, \delta) + \frac{\hat{\mathfrak{C}}_{\mathcal{S}}(\mathcal{F})}{C\pi_C} \sum_{y=1}^C \alpha_y \tilde{\beta}_y \sqrt{\pi_y} \left[1 - softmax\left(\beta_y B_y(f) + \Delta_y\right)\right]. \qquad (16)$$

From Eq.(16), we have the following insights, whose empirical validation can be found in Sec.4.2.

**(In1) Why re-weighting and logit-adjustment are necessary?** Due to the term $\sqrt{\pi_y}$ and $B_y(f)$, the generalization bound is also imbalanced among classes. Both re-weighting and logit-adjustment can obtain a sharper generalization bound by assigning different weights to the classes with different $\sqrt{\pi_y}$ and $B_y(f)$. In this process, $\alpha_y$ mainly rebalances the generalization performance among classes, *i.e.*, $\sqrt{\pi_y}$, while $\beta_y$ and $\Delta_y$ focus on adjusting the imbalance of the terms $B_y(f)$ among classes.

**(In2) Why the deferred scheme is necessary?** As pointed out in [11, 17], weighting up the minority classes will cause difficulties and instability in optimization, especially when the distribution is extremely imbalanced. To fix this issue, Cao et al. [12] develops a deferred scheme, where $\alpha_y = 1$ and $(1-p)/(1-p^{N_y}), p \in (0,1)$ during the initial and terminal phase of training, respectively. Although this scheme shows significant improvement, there is still a lack of theoretical explanation.

Fortunately, Prop.3 can give us some inspiration. Specifically, although a weighted loss can boost the optimization on the minority classes, it is harmful to the further improvement on the majority classes, as shown in Fig.3. Hence, the majority/minority classes will have relatively small/large $B_y(f)$ respectively, and the generalization bound becomes even looser. By contrast, in the DRW scheme, we have $\alpha_y = 1$ during the initial phase of training. Such a warm-up phase will encourage the model to focus on the majority classes and induce a small $B_y(f)$ for both majority and minority classes after weighting up the minority classes. On top of this, the generalization bound can become sharper, which explains the effectiveness of the deferred scheme.

**(In3) How does our result explain the design of existing losses?** On one hand, for re-weighting losses, $\alpha_y$ should decrease as $\pi_y$ increases, which is consistent with the balanced loss with $\alpha_y = \pi_y^{-1}$ [10] and $\alpha_y = (1-p)/(1-p^{N_y}), p \in (0,1)$ [11]. On the other hand, from the insight **(In2)**, we know that when $\alpha_y = 1$, $B_y(f)$ will be increasing *w.r.t.* $\pi_y$. Hence, for logit-adjustment losses, both $\beta_y$ and $\Delta_y$ should increase as $\pi_y$ increases. This insight is consistent with the LDAM loss ($\Delta_y \propto -N_y^{-1/4}$) [12], the logit-adjusted loss ($\Delta_y = \tau \log \pi_y$) [13], and the CDT loss ($\beta_y = (N_y/N_1)^{\gamma}$) [14].

**(In4) Are re-weigting and logit-adjustment fully compatible?** **(a)** Unfortunately, the answer is negative. To be specific, the re-weighting term $\alpha_y$ is decreasing *w.r.t.* $\pi_y$, whereas the multiplicative logit-adjustment term $\beta_y$ is increasing *w.r.t.* $\pi_y$. As a result, $\tilde{\beta}_y$ will weaken the effect of $\alpha_y$. **(b)** Fortunately, $\alpha_y$ is compatible with the additive logit-adjustment term $\Delta_y$ since both terms can induce a sharper generalization bound.

### 3.3 Principled Learning Algorithm induced by the Theoretical Insights

In this part, we present a principled learning algorithm induced by the theoretical insights in Sec.3.2. First, according to **(In1)-(In3)**, it is crucial to comprehensively utilize re-weighting, logit-adjustment, and the DRW scheme, as they all contribute to improving the generalization bound. Second, according to **(In4)**, we propose a Truncated Logit-Adjustment (TLA) scheme to avoid the conflict between $\alpha_y$ and $\beta_y$. In this scheme, $\beta_y$ still increases *w.r.t.* $\pi_y$ during the initial phase of training but is truncated to 1 during the terminal phase of training. Third, we set $\alpha_y \propto \pi_y^{-\nu}, \nu > 0$ to align $\alpha_y$ with $\sqrt{\pi_y}$, which we name Aligned DRW (ADRW). Note that such a re-weighting scheme also follows the Fisher consistency property presented in [13]. Finally, the overall algorithm is summarized in Alg.1, where the logit-adjustment methods mentioned in Sec.2.1 are all reasonable options for $\beta_y, \Delta_y$ in the line 5 and $\Delta_y$ in the line 7.

**Algorithm 1:** Principled Learning Algorithm induced by the Theoretical Insights

---

**Require:** Training set $\mathcal{S} = \{(x_i, y_i)\}_{i=1}^N$ and a model $f$ parameterized by $\Theta$.

---

1: Initialize the model parameters $\Theta$ randomly.

2: **for** $t = 1, 2, \cdots, T$ **do**

3:     $\mathcal{B} \leftarrow \text{SampleMiniBatch}(\mathcal{S}, m)$                             ▷ A mini-batch of $m$ samples

4:     **if** $t < T_0$ **then**

5:         Set $\alpha = 1, \beta_y, \Delta_y$                          ▷ Adjust logits during the initial phase

6:     **else**

7:         Set $\alpha_y \propto \pi_y^{-\nu}, \beta_y = 1, \Delta_y, \nu > 0$                ▷ TLA and ADRW

8:     **end if**

9:     $L(f, \mathcal{B}) \leftarrow \frac{1}{m} \sum_{(\boldsymbol{x},y) \in \mathcal{B}} L_{\text{VS}}(f(\boldsymbol{x}), y)$            ▷ Calculate the loss

10:    $\Theta \leftarrow \Theta - \eta \nabla_\Theta L(f, \mathcal{B})$                     ▷ One SGD step

11:    Optional: anneal the learning rate $\eta$.           ▷ Required when $t = T_0$

12: **end for**

---

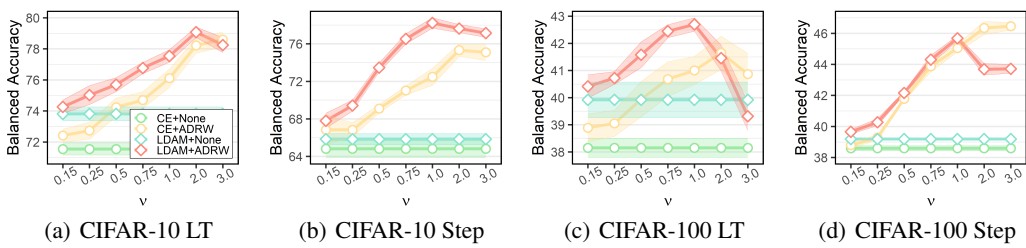

Figure 1: The balanced accuracy of the CE loss and the LDAM loss *w.r.t.* $\alpha_y \propto \pi_y^{-\nu}$ on the CIFAR datasets, where the imbalance ratio $\rho = 100$. Both re-weighting and logit-adjustment boost the model performance, which is consistent with the theoretical insight **(In1)** and **(In4-b)**.

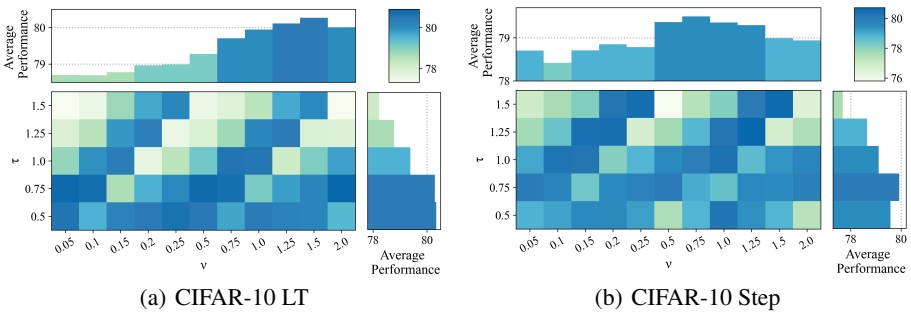

Figure 2: Sensitivity analysis of VS+ADRW *w.r.t.* $\alpha_y \propto \pi_y^{-\nu}$ and $\Delta_y = \tau \log \pi_y$ on the CIFAR-10 dataset, where the imbalance ratio $\rho = 100$. Both re-weighting and logit-adjustment boost the model performance, which is consistent with the theoretical insights **(In1)** and **(In4-b)**.

## 4 Experiments

### 4.1 Experiment Protocols

Here, we briefly introduce the experiment protocols, and more details can be found in Appendix F.

**Datasets.** We conduct the experiments on four popular benchmark datasets for imbalanced learning. **(a) CIFAR-10 and CIFAR-100**: Following the protocol in [26, 11, 12], we consider two types of imbalance: long-tailed imbalance (LT) and step imbalance (Step). For both imbalance types, we

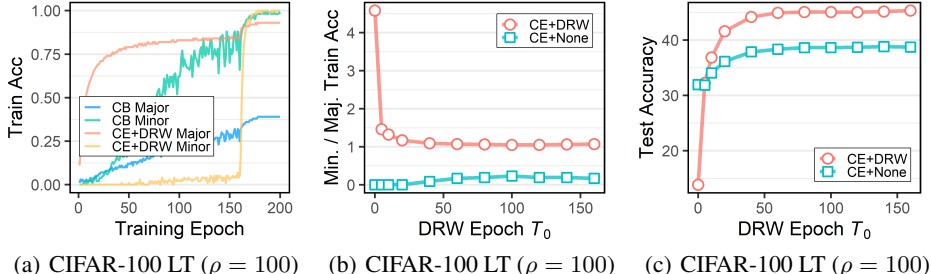

(a) CIFAR-100 LT ($\rho = 100$)     (b) CIFAR-100 LT ($\rho = 100$)     (c) CIFAR-100 LT ($\rho = 100$)

Figure 3: (a) Training accuracy of CE+DRW ($T_0 = 160$) and the CB loss *w.r.t.* training epoch. (b) $\widehat{Acc}_{\min}/\widehat{Acc}_{\max}$ *w.r.t.* the DRW epoch $T_0$, where $\widehat{Acc}_{\min}$ and $\widehat{Acc}_{\max}$ denote the training accuracy of the best model on the minority/majority classes, respectively. (c) The test accuracy of the best model *w.r.t.* the DRW epoch $T_0$. We can find that the DRW scheme balances the training accuracy between the majority classes and the minority classes and thus improves the model performance on the test set, which is consistent with the theoretical insight **(In2)**.

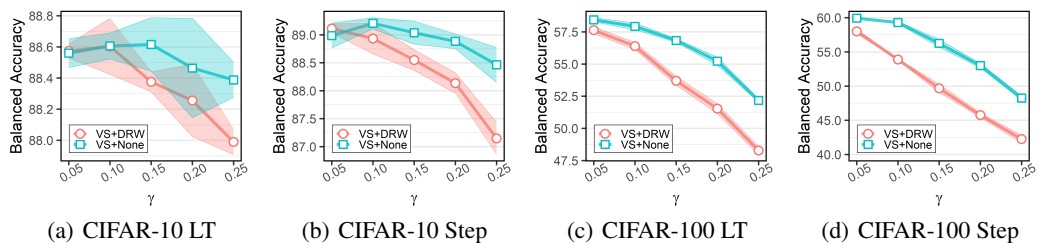

(a) CIFAR-10 LT     (b) CIFAR-10 Step     (c) CIFAR-100 LT     (d) CIFAR-100 Step

Figure 4: The balanced accuracy of the VS loss *w.r.t.* $\beta_y = (N_y/N_1)^\gamma$ on the CIFAR datasets, where the imbalance ratio $\rho = 10$. We can find that VS+DRW performs inferior to VS+None, especially when $\gamma$ is large, which is consistent with the theoretical insight **(In4-a).**

report the balanced accuracy averaged over 5 random seeds with an imbalance ratio $\rho := N_1/N_C \in \{10, 100\}$. **(b) ImageNet-LT and iNaturalist**: We use the long-tailed version of the ImageNet dataset[2] [2] proposed by [27], and iNaturalist[3] [5] is a real-world long-tailed dataset.

**Baselines and Competitors.** For the CIFAR datasets, we aim to validate the theoretical results and the performance gain induced by the proposed method. Hence, we select the following baselines: the CE loss (CE) [19], the LDAM loss (LDAM) [12], LDAM+DRW [12], and the VS loss (VS) [16] that generalizes the LA loss [13] and the CDT loss [14]. We tune all the hyperparameters according to the suggestions in the original papers. For the ImageNet-LT and iNaturalist datasets, we select state-of-the-art methods, listed in Tab.2, as the competitors to validate the effectiveness of the method.

**Implementation Details.** We implement three instances of the proposed learning algorithm: the CE loss equipped with the ADRW scheme (**CE+ADRW**), the LDAM loss equipped with the ADRW scheme (**LDAM+ADRW**), and the VS loss equipped with the TLA and the ADRW scheme (**VS+TLA+ADRW**). We tune the hyperparameter $\nu$, and the other hyperparameters follow those used in the baselines. In addition, we incorporate the Sharpness-Aware Minimization (SAM) technique [28] to facilitate the optimization of the minority classes, allowing them to escape saddle points and converge to flat minima [29].

## 4.2 Theory Validation

In this part, we aim to validate our theoretical insights presented in Sec.3.2 on the CIFAR datasets. Some more empirical results can be found in Appendix G.

---

[1]https://www.cs.toronto.edu/~kriz/cifar.html. Licensed MIT.

[2]https://image-net.org/index.php. Licensed MIT.

[3]https://github.com/visipedia/inat_comp/tree/master/2017. Licensed MIT.

Table 1: The balanced accuracy averaged over 5 random seeds on the CIFAR datasets. The best and the runner-up method for each protocol are marked with **red** and **blue**, respectively. The best baseline model is marked with underline.

| Dataset | CIFAR-10 | | | | CIFAR-100 | | | |
|---|---|---|---|---|---|---|---|---|
| Imbalance Type | LT | | Step | | LT | | Step | |
| Imbalance Ratio | 100 | 10 | 100 | 10 | 100 | 10 | 100 | 10 |
| w/o SAM | | | | | | | | |
| CE | $71.5_{\pm 0.4}$ | $87.0_{\pm 0.2}$ | $64.8_{\pm 0.9}$ | $85.1_{\pm 0.3}$ | $38.3_{\pm 0.4}$ | $56.7_{\pm 0.4}$ | $38.6_{\pm 0.2}$ | $54.4_{\pm 0.3}$ |
| LDAM | $73.8_{\pm 0.4}$ | $86.4_{\pm 0.4}$ | $65.8_{\pm 0.6}$ | $85.0_{\pm 0.3}$ | $39.9_{\pm 0.7}$ | $55.7_{\pm 0.5}$ | $39.2_{\pm 0.0}$ | $50.5_{\pm 0.2}$ |
| VS | $78.8_{\pm 0.2}$ | $\underline{88.7_{\pm 0.1}}$ | $76.1_{\pm 0.7}$ | $88.3_{\pm 0.1}$ | $41.8_{\pm 0.7}$ | $58.4_{\pm 0.2}$ | $\underline{46.2_{\pm 0.3}}$ | $\underline{59.9_{\pm 0.2}}$ |
| CE+DRW | $75.8_{\pm 0.3}$ | $87.9_{\pm 0.3}$ | $72.2_{\pm 0.8}$ | $88.0_{\pm 0.3}$ | $40.8_{\pm 0.6}$ | $58.1_{\pm 0.3}$ | $45.4_{\pm 0.4}$ | $59.1_{\pm 0.3}$ |
| LDAM+DRW | $77.7_{\pm 0.4}$ | $87.5_{\pm 0.2}$ | $77.8_{\pm 0.5}$ | $87.8_{\pm 0.3}$ | $42.7_{\pm 0.5}$ | $57.5_{\pm 0.3}$ | $45.3_{\pm 0.6}$ | $56.9_{\pm 0.2}$ |
| VS+DRW | $\underline{80.1_{\pm 0.1}}$ | $88.6_{\pm 0.1}$ | $78.2_{\pm 0.2}$ | $88.1_{\pm 0.1}$ | $\underline{41.3_{\pm 0.4}}$ | $57.6_{\pm 0.3}$ | $44.0_{\pm 0.3}$ | $58.0_{\pm 0.3}$ |
| **CE+ADRW** | $78.6_{\pm 0.5}$ | $88.2_{\pm 0.3}$ | $75.5_{\pm 0.6}$ | $88.5_{\pm 0.2}$ | $41.8_{\pm 0.6}$ | $58.3_{\pm 0.4}$ | $46.5_{\pm 0.3}$ | $59.2_{\pm 0.3}$ |
| **LDAM+ADRW** | $79.1_{\pm 0.2}$ | $87.6_{\pm 0.2}$ | $78.5_{\pm 0.4}$ | $88.1_{\pm 0.2}$ | $\textcolor{red}{43.0_{\pm 0.2}}$ | $58.0_{\pm 0.1}$ | $45.8_{\pm 0.2}$ | $57.6_{\pm 0.3}$ |
| **VS+TLA+DRW** | $\textcolor{blue}{80.8_{\pm 0.2}}$ | $\textcolor{blue}{88.8_{\pm 0.1}}$ | $\textcolor{blue}{80.0_{\pm 0.1}}$ | $\textcolor{blue}{89.2_{\pm 0.1}}$ | $43.0_{\pm 0.4}$ | $\textcolor{blue}{58.9_{\pm 0.1}}$ | $\textcolor{blue}{46.8_{\pm 0.1}}$ | $\textcolor{blue}{60.0_{\pm 0.3}}$ |
| **VS+TLA+ADRW** | $\textcolor{red}{81.1_{\pm 0.2}}$ | $\textcolor{red}{89.0_{\pm 0.2}}$ | $\textcolor{red}{80.9_{\pm 0.2}}$ | $\textcolor{red}{89.3_{\pm 0.1}}$ | $\textcolor{red}{43.4_{\pm 0.6}}$ | $\textcolor{red}{59.2_{\pm 0.2}}$ | $\textcolor{red}{47.8_{\pm 0.1}}$ | $\textcolor{red}{60.5_{\pm 0.3}}$ |
| w/ SAM | | | | | | | | |
| CE+DRW | $80.5_{\pm 0.2}$ | $89.8_{\pm 0.2}$ | $79.5_{\pm 0.3}$ | $90.2_{\pm 0.2}$ | $44.7_{\pm 0.6}$ | $60.7_{\pm 0.4}$ | $48.5_{\pm 0.3}$ | $61.7_{\pm 0.2}$ |
| LDAM+DRW | $81.6_{\pm 0.2}$ | $89.4_{\pm 0.2}$ | $81.2_{\pm 0.7}$ | $89.4_{\pm 0.1}$ | $45.2_{\pm 0.3}$ | $59.9_{\pm 0.2}$ | $49.1_{\pm 0.2}$ | $\underline{59.3_{\pm 0.2}}$ |
| VS | $82.6_{\pm 0.2}$ | $90.0_{\pm 0.1}$ | $83.2_{\pm 0.4}$ | $90.5_{\pm 0.1}$ | $45.9_{\pm 0.3}$ | $61.0_{\pm 0.3}$ | $\underline{47.4_{\pm 0.3}}$ | $61.6_{\pm 0.3}$ |
| **CE+ADRW** | $82.6_{\pm 0.2}$ | $\textcolor{blue}{90.1_{\pm 0.1}}$ | $\textcolor{blue}{82.8_{\pm 0.9}}$ | $90.2_{\pm 0.3}$ | $44.9_{\pm 0.6}$ | $\textcolor{red}{61.0_{\pm 0.4}}$ | $48.9_{\pm 0.2}$ | $\textcolor{blue}{62.1_{\pm 0.2}}$ |
| **LDAM+ADRW** | $\textcolor{blue}{83.0_{\pm 0.1}}$ | $89.7_{\pm 0.1}$ | $82.4_{\pm 0.3}$ | $90.0_{\pm 0.2}$ | $\textcolor{blue}{46.3_{\pm 0.4}}$ | $60.3_{\pm 0.3}$ | $\textcolor{red}{49.3_{\pm 0.4}}$ | $60.3_{\pm 0.2}$ |
| **VS+TLA+ADRW** | $\textcolor{red}{83.6_{\pm 0.2}}$ | $\textcolor{red}{90.3_{\pm 0.2}}$ | $\textcolor{red}{83.8_{\pm 0.1}}$ | $\textcolor{red}{90.8_{\pm 0.1}}$ | $\textcolor{red}{46.4_{\pm 0.6}}$ | $\textcolor{blue}{61.9_{\pm 0.3}}$ | $\textcolor{blue}{49.1_{\pm 0.2}}$ | $\textcolor{red}{62.3_{\pm 0.3}}$ |

**Validation of (In1) and (In4-b).** We report the model performance of the baselines *w.r.t.* the hyperparameters in Fig.1 and Fig.2. From these results, we can find that (1) Both CE+ADRW and LDAM perform better than CE. In other words, either re-weighting or logit-adjustment can boost the model performance. (2) LDAM+ADRW outperforms both CE+ADRW and LDAM, and for VS+ADRW, increasing the hyperparameters $\nu$ and $\tau$ appropriately can also bring performance gains. All these results validate the compatibility between re-weighting and the additive logit-adjustment.

**Validation of (In2).** We present a series of results in Fig.3, where the training accuracy of different classes represents the corresponding $B_f(y)$. To be specific, Fig.3(a) demonstrates the trend of training accuracy on the majority classes and the minority classes. For the CB loss, the learning process only focuses on the minority classes and hinders the performance improvement on the majority classes (CB Major *v.s.* CB Minor). Hence, an extremely imbalanced $B_y(f)$ induces a poor generalization performance. By contrast, the DRW scheme first focuses on the majority classes (CE+DRW Major *v.s.* CE+DRW Minor with the training epoch $t \leq T_0 = 160$) and then pays more attention to the minority classes during the terminal phase of training (CE+DRW Major *v.s.* CE+DRW Minor with the training epoch $t > T_0 = 160$). Benefiting from the DRW scheme, both the majority classes and the minority classes are well-trained and thus have a balanced term $B_y(f)$ (Fig.3(b)), leading to a corresponding improvement on the test accuracy (Fig.3(c)). Even if we remove the re-weighting term (CE+None), the imbalance degree of $B_y(f)$ is still consistent with the test performance. Note that the learning rate of CE+None is also decreased at the corresponding epoch $T_0$, making the line in Figure 3(b) and 3(c) not constant.

**Validation of (In4-a).** In Fig.4, we present the model performance of the VS loss *w.r.t.* the hyperparameter of the multiplicative logit-adjustment $\gamma$. We can find that VS+DRW performs inferior to VS, especially when $\gamma$ is large. This phenomenon validates the incompatibility between re-weighting and multiplicative logit-adjustment.

## 4.3 Performance Comparison

We report the empirical results on the CIFAR datasets in Tab.1, where the imbalance ratio $\rho \in \{10, 100\}$. From these results, we have the following observations: (1) The proposed learning algorithm consistently outperforms the baselines, especially when the dataset is more imbalanced.

Table 2: The balanced accuracy on the ImageNet-LT and iNaturalist datasets.

| Method | One stage | ImageNet-LT | | | | iNaturalist | | | |
|---|---|---|---|---|---|---|---|---|---|
| | | Many | Med. | Few | All | Many | Med. | Few | All |
| OLTR [27] | × | 43.2 | 35.1 | 18.5 | 35.6 | 59.0 | 64.1 | 64.9 | 63.9 |
| LFMR [30] | × | 47.1 | 35.0 | 17.5 | 37.2 | - | - | - | - |
| BBN [31] | × | - | - | - | - | 49.4 | 70.8 | 65.3 | 66.3 |
| cRT [32] | × | 61.8 | 46.2 | 27.3 | 49.6 | 69.0 | 66.0 | 63.2 | 65.2 |
| $\tau$-norm [32] | × | 59.1 | 46.9 | 30.7 | 49.4 | 65.6 | 65.3 | 65.5 | 65.6 |
| DiVE [33] | × | 64.1 | 50.4 | 31.5 | 53.1 | 70.6 | 70.0 | 67.6 | 69.1 |
| DisAlign [34] | × | 61.3 | 52.2 | 31.4 | 52.9 | 69.0 | **71.1** | 70.2 | 70.6 |
| WB [35] | × | 62.5 | 50.4 | **41.5** | 53.9 | 71.2 | 70.4 | 69.7 | 70.2 |
| CE [32] | ✓ | **65.9** | 37.5 | 7.7 | 44.4 | **72.2** | 63.0 | 57.2 | 61.7 |
| CE+CB [11] | ✓ | 39.6 | 32.7 | 16.8 | 33.2 | 53.4 | 54.8 | 53.2 | 54.0 |
| Focal [11] | ✓ | 36.4 | 29.9 | 16.0 | 30.5 | - | - | - | 61.1 |
| De-confound [36] | ✓ | 62.7 | 48.8 | 31.6 | 51.8 | - | - | - | - |
| DRO-LT [37] | ✓ | 64.0 | 49.8 | 33.1 | 53.5 | - | - | - | 69.7 |
| SAM [29] | ✓ | 62.0 | 52.1 | 34.8 | 53.1 | 64.1 | 70.5 | 71.2 | 70.1 |
| Ours | ✓ | 62.9 | **52.6** | 37.1 | **54.1** | 64.7 | 70.7 | **72.1** | **70.7** |

Such performance gains validate the effectiveness of the proposed methods (2) Both re-weighting and logit-adjustment can improve the model performance, which is consistent with the theoretical insight **(In1)** and **(In4-b)**. (3) When $\rho = 10$ or on the CIFAR-100 dataset, VS+DRW performs inferior to VS. Fortunately, when equipped with the proposed TLA scheme, VS+TLA+DRW outperforms both VS and VS+DRW. These results again validate our theoretical insight **(In4-a)**. (4) When $\rho = 10$, CE+ADRW outperforms LDAM+ADRW, and similar counter-intuitive phenomena are also observed in [29]. We conjecture that in this case, re-weighting is enough to rebalance the generalization bound, and the additional LDAM loss might induce other issues such as inconsistency.

We present the overall balanced accuracy on the ImageNet-LT and iNaturalist datasets in Tab.2, where SAM and Ours denotes LDAM+DRW+SAM and VS+TLA+ADRW+SAM, respectively. These results demonstrate that the proposed learning algorithm outperforms the competitors, especially the one-stage ones, which again confirms the effectiveness of the proposed learning algorithm.

# 5 Conclusion and Future Work

In this work, with the proposed local Lipschitz property and the data-dependent contraction technique, we present a unified generalization analysis of the loss-modification approaches for imbalanced learning. Benefiting from this fine-grained analysis, we not only reveal the role of both re-weighting and logit-adjustment approaches but also explain some empirical phenomena that are out of the scope of existing theories. Moreover, a principled learning algorithm is proposed based on the theoretical insights. Finally, extensive experimental results on benchmark datasets validate our theoretical analysis and the effectiveness of the proposed method.

Theoretically, one important future work is to provide a systematical Fisher consistency analysis for the VS loss, providing more insights to design re-weighting and logit-adjustment terms. Methodologically, it might be a promising direction to design an adaptive scheme that can automatically determine the hyperparameters of the learning algorithm.

# Acknowledgments

This work was supported in part by the National Key R&D Program of China under Grant 2018AAA0102000, in part by National Natural Science Foundation of China: 62236008, U21B2038, U2001202, 61931008, 62122075, 61976202 and 62206264, in part by the Fundamental Research Funds for the Central Universities, in part by Youth Innovation Promotion Association CAS, in part by the Strategic Priority Research Program of Chinese Academy of Sciences (Grant No. XDB28000000) and in part by the Innovation Funding of ICT, CAS under Grant No. E000000.

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
