| $71.5_{\pm0.4}$ | $87.0_{\pm0.2}$ | $64.8_{\pm0.9}$ | $85.1_{\pm0.3}$ | $38.3_{\pm0.4}$ | $56.7_{\pm0.4}$ | $38.6_{\pm0.2}$ | $54.4_{\pm0.3}$ |
| LDAM | $73.8_{\pm0.4}$ | $86.4_{\pm0.4}$ | $65.8_{\pm0.6}$ | $85.0_{\pm0.3}$ | $39.9_{\pm0.7}$ | $55.7_{\pm0.5}$ | $39.2_{\pm0.0}$ | $50.5_{\pm0.2}$ |
| VS | $78.8_{\pm0.2}$ | $\underline{88.7}_{\pm0.1}$ | $76.1_{\pm0.7}$ | $88.3_{\pm0.1}$ | $41.8_{\pm0.7}$ | $58.4_{\pm0.2}$ | $\underline{46.2}_{\pm0.3}$ | $\underline{59.9}_{\pm0.2}$ |
| CE+DRW | $75.8_{\pm0.3}$ | $87.9_{\pm0.3}$ | $72.2_{\pm0.8}$ | $88.0_{\pm0.3}$ | $40.8_{\pm0.6}$ | $58.1_{\pm0.3}$ | $45.4_{\pm0.4}$ | $59.1_{\pm0.3}$ |
| LDAM+DRW | $77.7_{\pm0.4}$ | $87.5_{\pm0.2}$ | $77.8_{\pm0.5}$ | $87.8_{\pm0.3}$ | $42.7_{\pm0.5}$ | $57.5_{\pm0.3}$ | $45.3_{\pm0.6}$ | $56.9_{\pm0.2}$ |
| VS+DRW | $\underline{80.1}_{\pm0.1}$ | $88.6_{\pm0.1}$ | $78.2_{\pm0.2}$ | $88.1_{\pm0.1}$ | $\underline{41.3}_{\pm0.4}$ | $57.6_{\pm0.3}$ | $44.0_{\pm0.3}$ | $58.0_{\pm0.3}$ |
| **CE+ADRW** | $78.6_{\pm0.5}$ | $88.2_{\pm0.3}$ | $75.5_{\pm0.6}$ | $88.5_{\pm0.2}$ | $41.8_{\pm0.6}$ | $58.3_{\pm0.4}$ | $46.5_{\pm0.3}$ | $59.2_{\pm0.3}$ |
| **LDAM+ADRW** | $79.1_{\pm0.2}$ | $87.6_{\pm0.2}$ | $78.5_{\pm0.4}$ | $88.1_{\pm0.2}$ | $\textcolor{red}{43.0}_{\pm0.2}$ | $58.0_{\pm0.1}$ | $45.8_{\pm0.2}$ | $57.6_{\pm0.3}$ |
| **VS+TLA+DRW** | $\textcolor{blue}{80.8}_{\pm0.2}$ | $\textcolor{blue}{88.8}_{\pm0.1}$ | $\textcolor{blue}{80.0}_{\pm0.1}$ | $\textcolor{blue}{89.2}_{\pm0.1}$ | $43.0_{\pm0.4}$ | $\textcolor{blue}{58.9}_{\pm0.1}$ | $\textcolor{blue}{46.8}_{\pm0.1}$ | $\textcolor{blue}{60.0}_{\pm0.3}$ |
| **VS+TLA+ADRW** | $\textcolor{red}{81.1}_{\pm0.2}$ | $\textcolor{red}{89.0}_{\pm0.2}$ | $\textcolor{red}{80.9}_{\pm0.2}$ | $\textcolor{red}{89.3}_{\pm0.1}$ | $\textcolor{red}{43.4}_{\pm0.6}$ | $\textcolor{red}{59.2}_{\pm0.2}$ | $\textcolor{red}{47.8}_{\pm0.1}$ | $\textcolor{red}{60.5}_{\pm0.3}$ |
| w/ SAM | | | | | | | | |
| CE+DRW | $80.5_{\pm0.2}$ | $89.8_{\pm0.2}$ | $79.5_{\pm0.3}$ | $90.2_{\pm0.2}$ | $44.7_{\pm0.6}$ | $60.7_{\pm0.4}$ | $48.5_{\pm0.3}$ | $61.7_{\pm0.2}$ |
| LDAM+DRW | $81.6_{\pm0.2}$ | $89.4_{\pm0.2}$ | $81.2_{\pm0.7}$ | $89.4_{\pm0.1}$ | $45.2_{\pm0.3}$ | $59.9_{\pm0.2}$ | $49.1_{\pm0.2}$ | $\underline{59.3}_{\pm0.2}$ |
| VS | $82.6_{\pm0.2}$ | $90.0_{\pm0.1}$ | $83.2_{\pm0.4}$ | $90.5_{\pm0.1}$ | $45.9_{\pm0.3}$ | $61.0_{\pm0.3}$ | $\underline{47.4}_{\pm0.3}$ | $61.6_{\pm0.3}$ |
| **CE+ADRW** | $82.6_{\pm0.2}$ | $\textcolor{blue}{90.1}_{\pm0.1}$ | $\textcolor{blue}{82.8}_{\pm0.9}$ | $90.2_{\pm0.3}$ | $44.9_{\pm0.6}$ | $\textcolor{red}{61.0}_{\pm0.4}$ | $48.9_{\pm0.2}$ | $\textcolor{blue}{62.1}_{\pm0.2}$ |
| **LDAM+ADRW** | $\textcolor{blue}{83.0}_{\pm0.1}$ | $89.7_{\pm0.1}$ | $82.4_{\pm0.3}$ | $90.0_{\pm0.2}$ | $\textcolor{blue}{46.3}_{\pm0.4}$ | $60.3_{\pm0.3}$ | $\textcolor{red}{49.3}_{\pm0.4}$ | $60.3_{\pm0.2}$ |
| **VS+TLA+ADRW** | $\textcolor{red}{83.6}_{\pm0.2}$ | $\textcolor{red}{90.3}_{\pm0.2}$ | $\textcolor{red}{83.8}_{\pm0.1}$ | $\textcolor{red}{90.8}_{\pm0.1}$ | $\textcolor{red}{46.4}_{\pm0.6}$ | $\textcolor{blue}{61.9}_{\pm0.3}$ | $\textcolor{blue}{49.1}_{\pm0.2}$ | $\textcolor{red}{62.3}_{\pm0.3}$ |

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

# Appendix

## A    Related Work

In the main text, we have discussed loss-oriented methods for imbalanced learning. Next, we will briefly review the other orthogonal directions in this field, and more details can be found in the latest survey [38].

**Data-oriented methods** aim to resample the training set to achieve a more balanced data distribution. For example, Kubat and Matwin [39] and Chawla et al. [40] over-sample the minority classes, and Mani and Zhang [41] and Drummond et al. [42] under-sample the majority classes. Although being intuitive, over-sampling might suffer from over-fiiting, and under-sampling reduces the total amount of information that the model can learn from [43].

**Post-hoc methods** follow a naïve training process but make adjustments during the test phase. For example, Collell et al. [44] calibrate the decision-making condition via data priors. Kang et al. [32] propose to balance the the classifier weights via $\tau$-normalization. Menon et al. [13] adjust the logits such that the predictions can align with the balanced accuracy.

**Decoupling methods** follow a two-stage learning paradigm. The first stage aims to learning features based on a naïve learning process. At the second stage, Kang et al. [32] retrain the classifier under a balanced label distribution; Alshammari et al. [35] exploit L2-normalization, weight decay, and maxNorm constraint to achieve balanced classifier weights.

**Ensemble methods** aggregate multiple expert models trained on different data regimes. For example, Cai et al. [45] distribute diverse but overlapping class splits for experts and encourage each expert to learn complementary knowledge. Wang et al. [46] propose a dynamic routing framework that reduces model variance and model bias to mitigate the performance degeneration on majority classes. Besides, Zhou et al. [31] propose a cumulative learning strategy that first learns the universal patterns and then pays attention to the minority classes gradually.

Besides, Tang et al. [36] propose a causal inference framework to remove unfavorable GSD momentum. Cui et al. [47] exploit contrastive learning to make full use of aggressive data augmentation techniques. Rangwani et al. [29] utilize Sharpness-Aware Minimization (SAM) technique to boost the optimization of the minority classes, allowing them to escape saddle points and converge to flat minima.

## B    Proof of the Basic Lemma (Lem.2)

**Lemma 2.** *Given the function set $\mathcal{F}$ and a loss function $L : \mathbb{R}^C \times \mathcal{Y} \to [0, M]$, then for any $\delta \in (0, 1)$, with probability at least $1 - \delta$ over the training set $\mathcal{S}$, the following generalization bound holds for all $g \in \mathcal{G}$:*

$$\mathcal{R}_{bal}^L(f) \precsim \Phi(L, \delta) + \frac{1}{C\pi_C} \cdot \hat{\mathfrak{C}}_{\mathcal{S}}(\mathcal{G}),  \tag{11}$$

*where $\Phi(L, \delta) := \frac{1}{C\pi_C}[\widehat{\mathcal{R}}^L(f) + 3M\sqrt{\frac{\log 2/\delta}{2N}}]$ contains the empirical risk on $\mathcal{S}$ and the $\delta$ term.*

*Proof.* On one hand,

$$\mathcal{R}^L(f) = \mathop{\mathbb{E}}_{(\boldsymbol{x},y)\sim\mathcal{D}}[L(f(\boldsymbol{x}), y)] = \sum_{y=1}^C \pi_y \mathop{\mathbb{E}}_{\boldsymbol{x}\sim\mathcal{D}_y}[L(f(\boldsymbol{x}), y)] = \sum_{j=1}^C \pi_y \mathcal{R}_y^L(f).  \tag{17}$$

On the other hand,

$$\mathcal{R}_{\text{bal}}^L(f) = \frac{1}{C}\sum_{j=1}^C \mathcal{R}_y^L(f) = \frac{1}{C}\sum_{j=1}^C \frac{1}{\pi_y} \cdot \pi_y \mathcal{R}_y^L(f) \leq \frac{1}{C\pi_C}\sum_{j=1}^C \pi_y \mathcal{R}_y^L(f) = \frac{1}{C\pi_C}\mathcal{R}^L(f),  \tag{18}$$

where the inequality comes from the fact that $\forall \boldsymbol{a}, \boldsymbol{b} \in \mathbb{R}^C, |\langle \boldsymbol{a}, \boldsymbol{b} \rangle| \leq \|\boldsymbol{a}\|_\infty \|\boldsymbol{b}\|_1$. Then, combining the traditional results in [19], for any $\delta \in (0, 1)$, with probability at least $1 - \delta$ over the training set

$\mathcal{S}$, the following generalization bound holds for all the $g \in \mathcal{G}$:

$$\mathcal{R}_{\text{bal}}^L(f) \precsim \frac{1}{C\pi_C} \left[ \widehat{\mathcal{R}}^L(f) + \hat{\mathfrak{C}}_{\mathcal{S}}(\mathcal{G}) + 3M\sqrt{\frac{\log 2/\delta}{2N}} \right]. \tag{19}$$

$\square$

## C  Proof of the Data-Dependent Contraction Lemma (Lem.3)

**Lemma 3** (Data-Dependent Contraction). *Assume that the loss function $L(f, \boldsymbol{x})$ is local Lipschitz continuous with constants $\{\mu_y\}_{y=1}^C$. Then, the following inequality holds under Asm.1:*

$$\hat{\mathfrak{C}}_{\mathcal{S}}(\mathcal{G}) \precsim \hat{\mathfrak{C}}_{\mathcal{S}}(\mathcal{F}) \sum_{y=1}^C \mu_y \sqrt{\pi_y}, \tag{13}$$

*Proof.* According to the definition of complexity, we have

$$\begin{aligned}
\hat{\mathfrak{C}}_{\mathcal{S}}(\mathcal{G}) &= \mathbb{E}_{\boldsymbol{\xi}} \left[ \sup_{g \in \mathcal{G}} \frac{1}{N} \sum_{n=1}^N \xi^{(n)} g(\boldsymbol{z}^{(n)}) \right] = \mathbb{E}_{\boldsymbol{\xi}} \left[ \sup_{g \in \mathcal{G}} \frac{1}{N} \sum_{y=1}^C \sum_{n=1}^{N_y} \xi_y^{(n)} g(\boldsymbol{z}_y^{(n)}) \right] \\
&\leq \sum_{y=1}^C \mathbb{E}_{\boldsymbol{\xi}_y} \left[ \frac{1}{N} \sup_{g \in \mathcal{G}} \sum_{n=1}^{N_y} \xi_y^{(n)} g(\boldsymbol{z}_y^{(n)}) \right] = \sum_{y=1}^C \frac{N_y}{N} \mathbb{E}_{\boldsymbol{\xi}_y} \left[ \frac{1}{N_y} \sup_{g \in \mathcal{G}} \sum_{n=1}^{N_y} \xi_y^{(n)} g(\boldsymbol{z}_y^{(n)}) \right] \\
&= \sum_{y=1}^C \pi_y \hat{\mathfrak{C}}_{\mathcal{S}_y}(\mathcal{G}) \precsim \sum_{y=1}^C \sqrt{\pi_y} \mu_y \hat{\mathfrak{C}}_{\mathcal{S}}(\mathcal{F}),
\end{aligned} \tag{20}$$

where the last inequality comes from Asm.1. $\square$

## D  Proof of Prop.2

**Proposition 2.** *Assume that $\mu_y \propto N_y^{-\kappa}, \kappa > 0$. Then, when $\kappa > 1$, the data-dependent bound presented in Thm.1 is sharper than the union bound defined in Prop.1.*

*Proof.* According to Def.1, it is not difficult to obtain $\mu = \max_y \mu_y \propto N_C^{-\kappa}$. Then, for the union bound, we have

$$\frac{\mu}{C} \sum_{y=1}^C \hat{\mathfrak{C}}_{\mathcal{S}_y}(\mathcal{F}) \propto \frac{\mu \cdot \hat{\mathfrak{C}}_{\mathcal{S}}(\mathcal{F})}{C} \sum_{y=1}^C \pi_y^{-0.5} \propto \frac{\hat{\mathfrak{C}}_{\mathcal{S}}(\mathcal{F})}{CN_C^{\kappa}} \sum_{y=1}^C \pi_y^{-0.5} = \frac{\hat{\mathfrak{C}}_{\mathcal{S}}(\mathcal{F})}{CN^{\kappa}\pi_C^{\kappa}} \sum_{y=1}^C \pi_y^{-0.5}, \tag{21}$$

where the first equality comes from Asm.1. For the data-dependent bound, we have

$$\frac{\hat{\mathfrak{C}}_{\mathcal{S}}(\mathcal{F})}{C\pi_C} \sum_{y=1}^C \mu_y \sqrt{\pi_y} \propto \frac{\hat{\mathfrak{C}}_{\mathcal{S}}(\mathcal{F})}{C\pi_C} \sum_{y=1}^C N_y^{-\kappa} \sqrt{\pi_y} = \frac{\hat{\mathfrak{C}}_{\mathcal{S}}(\mathcal{F})}{CN^{\kappa}\pi_C} \sum_{y=1}^C \pi_y^{0.5-\kappa}. \tag{22}$$

Let

$$h_1(\kappa) := \frac{1}{\pi_C^{\kappa}} \sum_{y=1}^C \pi_y^{-0.5} - \frac{1}{\pi_C} \sum_{y=1}^C \pi_y^{0.5-\kappa}. \tag{23}$$

Then, we have

$$\begin{aligned}
h_1'(\kappa) &= -\ln \pi_C \cdot \frac{1}{\pi_C^{\kappa}} \sum_{y=1}^C \pi_y^{-0.5} + \frac{1}{\pi_C} \sum_{y=1}^C \pi_y^{0.5-\kappa} \ln \pi_y \\
&= -\ln \pi_C \cdot \frac{1}{\pi_C^{\kappa}} \sum_{y=1}^C \pi_y^{-0.5} + \frac{1}{\pi_C} \sum_{y=1}^C \pi_y^{-0.5} \underbrace{\pi_y^{1-\kappa} \ln \pi_y}_{h_2(\pi_y)}.
\end{aligned} \tag{24}$$

Then, let $h_2(t) = t^{1-\kappa} \ln t, t \in (0, 1)$. When $\kappa > 1$, we have

$$h_2'(t) = (1 - \kappa)t^{-\kappa} \ln t + t^{1-\kappa} \cdot \frac{1}{t} = [(1 - \kappa) \ln t + 1] t^{-\kappa} > 0. \tag{25}$$

Thus, $\forall y \in \mathcal{Y}, h_2(\pi_y) \geq h_2(\pi_C)$. Then, we have

$$h_1'(\kappa) \geq -\ln \pi_C \cdot \frac{1}{\pi_C^\kappa} \sum_{y=1}^C \pi_y^{-0.5} + \frac{1}{\pi_C} \sum_{y=1}^C \pi_y^{-0.5} h_2(\pi_C) = 0. \tag{26}$$

Finally, the proof ends by the fact that $h_1(1) = 0$. $\qquad\square$

# E   Proof of the Local Lipschitz Property of the VS Loss (Prop.4)

**Lemma 5.** *Given* $\{a_i\}_{i=1}^C, \{b_i\}_{i=1}^C$, *if* $a_i, b_i \geq 0$, *we have* $\sum_{i=1}^C a_i^2 b_i^2 \leq \left(\sum_{i=1}^C a_i\right)^2 \left(\sum_{i=1}^C b_i\right)^2$.

*Proof.* According to the definition, we have

$$
\begin{aligned}
\left(\sum_{i=1}^C a_i\right)^2 \left(\sum_{i=1}^C b_i\right)^2 &= \left(\sum_{i=1}^C a_i^2 + 2\sum_{i \neq j} a_i a_j\right) \left(\sum_{i=1}^C b_i^2 + 2\sum_{i \neq j} b_i b_j\right) \\
&\geq \sum_{i=1}^C a_i^2 \cdot \sum_{i=1}^C b_i^2 = \sum_{i=1}^C a_i^2 b_i^2 + \sum_{i \neq j} a_j^2 b_j^2 \geq \sum_{i=1}^C a_i^2 b_i^2.
\end{aligned}
\tag{27}
$$

$\qquad\square$

**Lemma 4.** *Assume that the score function is bounded. Then, the VS loss is local Lipschitz continuous with constants* $\{\mu_y\}_{y=1}^C$, *where*

$$\mu_y = \alpha_y \tilde{\beta}_y \left[1 - \text{softmax}\left(\beta_y B_y(f) + \Delta_y\right)\right], \tag{15}$$

$\tilde{\beta}_y := \sqrt{\beta_y^2 + \left(\sum_{y' \neq y} \beta_{y'}\right)^2}$; *softmax* $(\cdot)$ *denotes the softmax function;* $B_y(f)$ *denotes the minimal prediction on the ground-truth class* $y$, *i.e.,* $B_y(f) := \min_{\mathbf{x} \in S_y} f(\boldsymbol{x})_y$.

*Proof.* According to the definition of the VS loss, we have

$$
\begin{aligned}
L_{\text{VS}}(f(\boldsymbol{x}), y) &= -\alpha_y \log\left(\frac{e^{\beta_y f(\boldsymbol{x})_y + \Delta_y}}{\sum_{y'} e^{\beta_{y'} f(\boldsymbol{x})_{y'} + \Delta_{y'}}}\right) \\
&= \alpha_y \log[1 + \sum_{y' \neq y} e^{\beta_{y'} f(\boldsymbol{x})_{y'} - \beta_y f(\boldsymbol{x})_y + \Delta_{y'} - \Delta_y}],
\end{aligned}
\tag{28}
$$

Let $\boldsymbol{s} := f(\boldsymbol{x})$, and define

$$\ell_y(\boldsymbol{s}) := \sum_{y' \neq y} e^{\beta_{y'} \boldsymbol{s}_{y'} + \Delta_{y'}}. \tag{29}$$

In other words, $L_{\text{VS}}(f, y) = \alpha_y \log\left[1 + e^{-(\beta_y \boldsymbol{s}_y + \Delta_y)} \ell_y(\boldsymbol{s})\right]$. Then,

$$
\begin{aligned}
\frac{\partial L_{\text{VS}}(f, y)}{\partial \boldsymbol{s}_y} &= -\alpha_y \beta_y \frac{e^{-(\beta_y \boldsymbol{s}_y + \Delta_y)} \ell_y(\boldsymbol{s})}{1 + e^{-(\beta_y \boldsymbol{s}_y + \Delta_y)} \ell_y(\boldsymbol{s})}, \\
\frac{\partial L_{\text{VS}}(f, y)}{\partial \boldsymbol{s}_{y'}} &= \alpha_y \beta_{y'} \frac{e^{-(\beta_y \boldsymbol{s}_y + \Delta_y)}}{1 + e^{-(\beta_y \boldsymbol{s}_y + \Delta_y)} \ell_y(\boldsymbol{s})} \cdot e^{\beta_{y'} \boldsymbol{s}_{y'} + \Delta_{y'}}, y' \neq y.
\end{aligned}
\tag{30}
$$

Hence,

$$\|\nabla_{\boldsymbol{s}} L_{\mathrm{VS}}(f, y)\|^2 = \left[\beta_y^2 \ell_y(\boldsymbol{s})^2 + \sum_{y' \neq y} \left(\beta_{y'} e^{\beta_{y'} \boldsymbol{s}_{y'} + \Delta_{y'}}\right)^2\right] \cdot \left[\frac{\alpha_y e^{-(\beta_y \boldsymbol{s}_y + \Delta_y)}}{1 + e^{-(\beta_y \boldsymbol{s}_y + \Delta_y)} \ell_y(\boldsymbol{s})}\right]^2$$

$$\leq \left[\beta_y^2 \ell_y(\boldsymbol{s})^2 + \left(\sum_{y' \neq y} \beta_{y'}\right)^2 \left(\sum_{y' \neq y} e^{\beta_{y'} \boldsymbol{s}_{y'} + \Delta_{y'}}\right)^2\right] \cdot \left[\frac{\alpha_y e^{-(\beta_y \boldsymbol{s}_y + \Delta_y)}}{1 + e^{-(\beta_y \boldsymbol{s}_y + \Delta_y)} \ell_y(\boldsymbol{s})}\right]^2 \qquad (31)$$

$$= \left[\beta_y^2 + \left(\sum_{y' \neq y} \beta_{y'}\right)^2\right] \cdot \left[\frac{\alpha_y e^{-(\beta_y \boldsymbol{s}_y + \Delta_y)} \ell_y(\boldsymbol{s})}{1 + e^{-(\beta_y \boldsymbol{s}_y + \Delta_y)} \ell_y(\boldsymbol{s})}\right]^2,$$

where the inequality comes from Lem.5. Thus,

$$\|\nabla_{\boldsymbol{s}} L_{\mathrm{VS}}(f, y)\| \leq \alpha_y \sqrt{\beta_y^2 + \left(\sum_{y' \neq y} \beta_{y'}\right)^2} \frac{e^{-(\beta_y \boldsymbol{s}_y + \Delta_y)} \ell_y(\boldsymbol{s})}{1 + e^{-(\beta_y \boldsymbol{s}_y + \Delta_y)} \ell_y(\boldsymbol{s})}$$

$$= \alpha_y \sqrt{\beta_y^2 + \left(\sum_{y' \neq y} \beta_{y'}\right)^2} \frac{\ell_y(\boldsymbol{s})}{e^{\beta_y \boldsymbol{s}_y + \Delta_y} + \ell_y(\boldsymbol{s})}$$

$$= \alpha_y \sqrt{\beta_y^2 + \left(\sum_{y' \neq y} \beta_{y'}\right)^2} \left[1 - \frac{e^{\beta_y \boldsymbol{s}_y + \Delta_y}}{\sum_{y'} e^{\beta_{y'} \boldsymbol{s}_{y'} + \Delta_{y'}}}\right] \qquad (32)$$

$$= \alpha_y \sqrt{\beta_y^2 + \left(\sum_{y' \neq y} \beta_{y'}\right)^2} \left[1 - softmax\left(\beta_y \boldsymbol{s}_y + \Delta_y\right)\right]$$

Since the score function is bounded, for any $y \in \mathcal{Y}$, there exists a constant $B_y(f)$ such that $B_y(f) = \inf_{\boldsymbol{x} \in \mathcal{S}_y} \boldsymbol{s}_y$, which completes the proof. □

# F  More Experiment Protocols

**Datasets.** We conduct experiments on four popular benchmark datasets for imbalanced learning. **(a) CIFAR-10 LT and CIFAR-100 LT**: The original version of CIFAR-10[1] and CIFAR-100[1] [1] consists of 50,000 training images and 10,000 validation images, uniformly sampled from 10 and 100 classes, respectively. Following the protocol in [26, 11, 12], we consider two types of imbalance: long-tailed imbalance, where the number of training samples for each class decreases exponentially, and step imbalance, which reduces the sample size of half of the classes to a fixed ratio. Then, let $\rho := N_1/N_C$ denote the imbalance ratio. For both imbalance types, we report the balanced accuracy averaged over 5 random seeds with $\rho \in \{10, 100\}$. **(b) ImageNet-LT and iNaturalist**: We use the long-tailed version of the ImageNet dataset[2] [2] proposed by [27], which contains 115.8K images from 1K classes with $N_1 = 1280, N_C = 5$. iNaturalist[3] [5] is a real-world long-tailed dataset with 437.5K images from 8,142 classes. Following the protocol in [48], the classes are split into three subsets under long-tailed imbalance: Head, Medium, and Tail. We report the balanced accuracy on all the classes and each subset.

**Backbones and Optimization Methods.** For the CIFAR datasets, we follow the implementation in [12]. Specifically, we train the ResNet-32 model [49] for 200 epochs by SGD with a momentum of 0.9, a weight decay of 2e-4, and a bath size of 128 [50]. A multistep learning rate schedule is used with an initial learning rate of 0.1, divided by 10 at the 160th and 180th epoch by default. For the

---
[1] https://www.cs.toronto.edu/~kriz/cifar.html. Licensed MIT.

[2] https://image-net.org/index.php. Licensed MIT.

[3] https://github.com/visipedia/inat_comp/tree/master/2017. Licensed MIT.

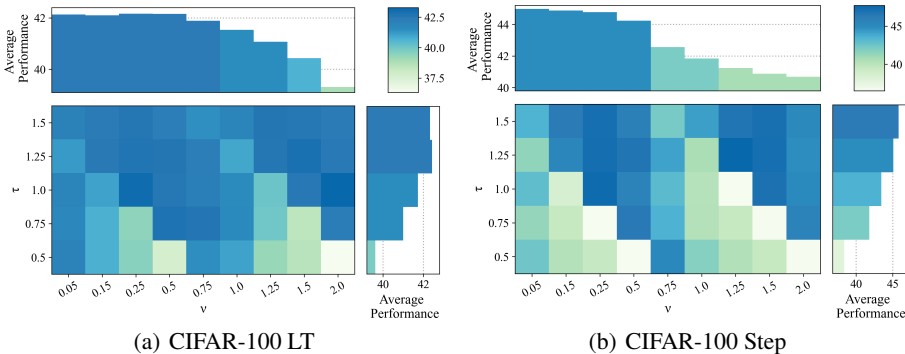

|             |                |
| ----------- | -------------- |
| (a) CIFAR-100 LT | (b) CIFAR-100 Step |

Figure 5: Sensitivity analysis of VS+ADRW *w.r.t.* $\alpha_y \propto \pi_y^{-\nu}$ and $\Delta_y = \tau \log \pi_y$ on the CIFAR datasets, where the imbalance ratio $\rho = 100$. Both re-weighting and logit-adjustment boost the model performance, which is consistent with the theoretical insights **(In1)** and **(In4-b)**.

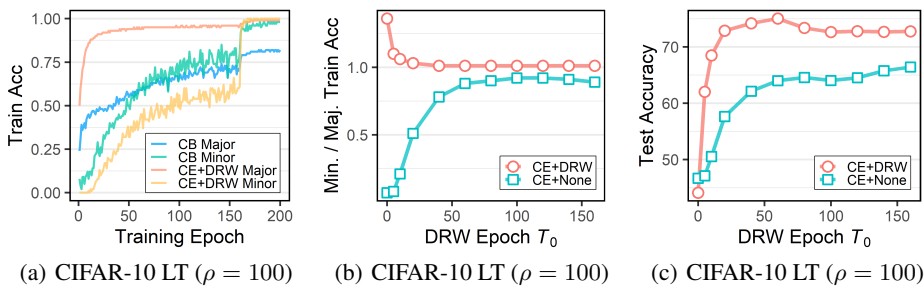

| (a) CIFAR-10 LT ($\rho = 100$) | (b) CIFAR-10 LT ($\rho = 100$) | (c) CIFAR-10 LT ($\rho = 100$) |
| --- | --- | --- |

Figure 6: (a) Training accuracy of CE+DRW ($T_0 = 160$) and the CB loss ($\alpha_y = (1-p)/(1-p^{N_y})$). (b) The ratio of the training accuracy between the minority classes and the majority classes of the best model *w.r.t.* the DRW epoch $T_0$. (c) The test accuracy of the best model *w.r.t.* the DRW epoch $T_0$. We can find that the DRW scheme balances the training accuracy between the majority classes and the minority classes and thus improves the model performance on the test set, which is consistent with the theoretical insight **(In2)**.

ImageNet-LT and iNaturalist datasets, we follow the implementation in [48, 29], where the ResNet-50 model [49] is trained for 90 epochs by SGD with a momentum of 0.9, a weight decay of 2e-4, and a batch size of 256. A cosine learning rate schedule is used with an initial learning rate of 0.1 and 0.2 for ImageNet-LT and iNaturalist, respectively. In addition, we incorporate the Sharpness-Aware Minimization (SAM) technique [28, 29] to facilitate the optimization of the minority classes, allowing them to escape saddle points and converge to flat minima. And the hyperparameter of SAM is tuned as suggested in [29].

**Infrastructure.** The experiments on the CIFAR datasets are carried out on an Ubuntu server equipped with Nvidia(R) RTX 3090 GPUs, whereas the experiments on ImageNet-LT and iNaturalist are conducted on NVIDIA(R) A100 GPUs. We implement the codes via `python` (v-3.8.10), and the main third-party packages include `pytorch` (v-1.8.0) [51], `numpy` (v-1.20.2) [52], `scikit-learn` (v-1.0.2) [53] and `torchvision` (v-0.9.0) [54].

**Parameter search.** We first tune the parameters via grid search according to the results in prior arts [13, 16]. To be specific, $\alpha_y \propto \pi_y^{-\nu}$, and $\nu$ is searched in $\{0.15, 0.25, 0.75, 1.0, 2.0, 3.0\}$; $\Delta_y = \tau \log \pi_y$, and $\tau$ is searched in $\{0.5, 0.75, 1.0, 1.25, 2.0\}$; $\beta_y = (N_y/N_1)^\gamma$, and $\gamma$ is searched in $\{0.05, 0.1, 0.15, 0.2, 0.25\}$. At first glance, the search space is relatively large. However, benefiting from the theoretical validation presented in Sec.4.2, the time complexity can be significantly decreased later. For example, according to Fig.4, we will choose a small $\gamma$ when $\nu$ is large to avoid the incompatibility issue.

# G  More Empirical Results

## G.1  More Empirical Results for Theory Validation

We present the sensitivity analysis of VS+ADRW on the CIFAR-100 dataset in Fig.5, where the imbalance ratio $\rho = 100$. Similar to the results on the CIFAR-10 dataset, appropriately increasing $\nu$ and $\tau$ can improve the model performance, which again validates the theoretical insights **(In1)** and **(In4-b)**.

Similar to Fig.3, Fig.6 provides a series of results on the CIFAR-10 dataset to validate the theoretical insight **(In2)**. Once again, the imbalance of $B_y(f)$ is highly correlated with the model performance on the test set, not only for CE+DRW but also for CE+None. It is worth mentioning that the optimal DRW epoch $T_0$ is 60, which is different from the commonly chosen value of 160. This might be because the CIFAR-10 dataset only has 10 classes, and a small DRW epoch $T_0$ may lead to overfitting on the majority classes.

In Fig.7-Fig.9, we provide the results under the imbalance ratio $\rho = 10$. The results are similar to those with $\rho = 100$, which again validates our theoretical results.

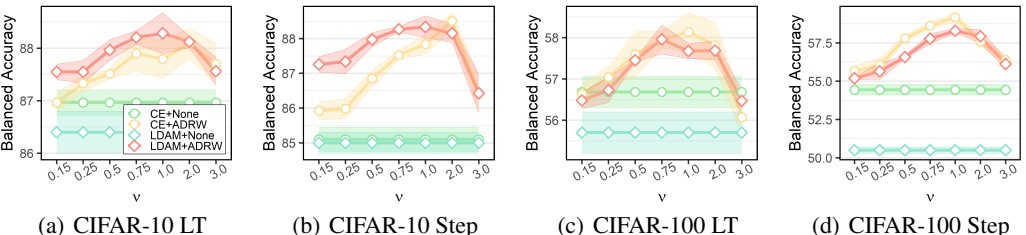

Figure 7: The balanced accuracy of the CE loss and the LDAM loss *w.r.t.* $\alpha_y \propto \pi_y^{-\nu}$ on the CIFAR datasets, where the imbalance ratio $\rho = 10$. Both re-weighting and logit-adjustment boost the model performance, which is consistent with the theoretical insight **(In1)** and **(In4-b)**.

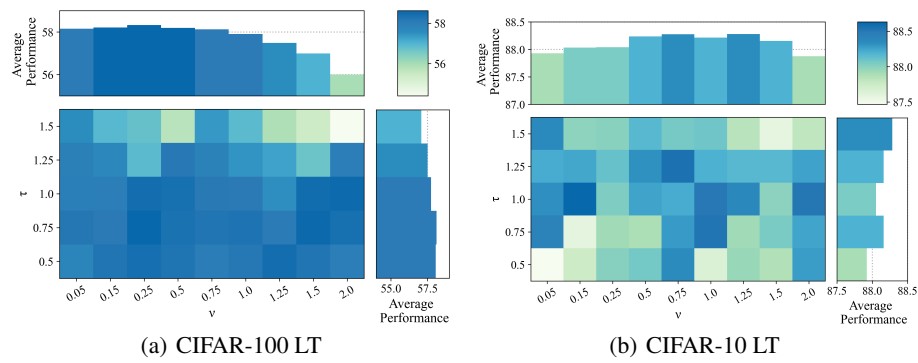

Figure 8: Sensitivity analysis of VS+ADRW *w.r.t.* $\alpha_y \propto \pi_y^{-\nu}$ and $\Delta_y = \tau \log \pi_y$ on the CIFAR datasets, where the imbalance ratio $\rho = 10$. Both re-weighting and logit-adjustment boost the model performance, which is consistent with the theoretical insights **(In1)** and **(In4-b)**.

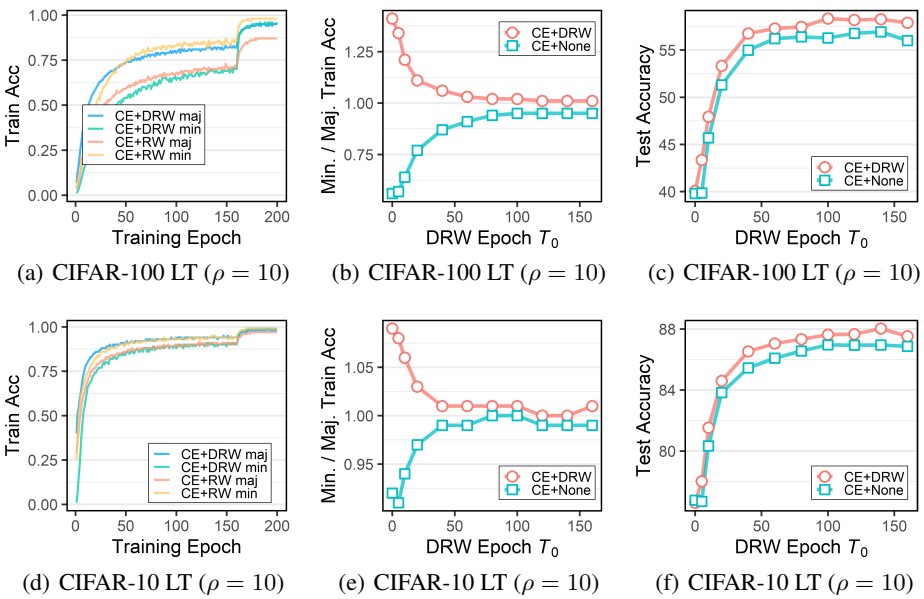

(a) CIFAR-100 LT ($\rho = 10$)     (b) CIFAR-100 LT ($\rho = 10$)     (c) CIFAR-100 LT ($\rho = 10$)

(d) CIFAR-10 LT ($\rho = 10$)     (e) CIFAR-10 LT ($\rho = 10$)     (f) CIFAR-10 LT ($\rho = 10$)

Figure 9: **(a, d)** Training accuracy of CE+DRW ($T_0 = 160$) and the CB loss *w.r.t.* training epoch. **(b, e)** $\widehat{Acc}_{\min}/\widehat{Acc}_{\text{maj}}$ *w.r.t.* the DRW epoch $T_0$, where $\widehat{Acc}_{\min}$ and $\widehat{Acc}_{\text{maj}}$ denote the training accuracy of the best model on the minority/majority classes, respectively. **(c, f)** The test accuracy of the best model *w.r.t.* the DRW epoch $T_0$. We can find that the DRW scheme balances the training accuracy between the majority classes and the minority classes and thus improves the model performance on the test set, which is consistent with the theoretical insight **(In2)**.