# OpenReview forum: "A Unified Generalization Analysis of Re-Weighting and Logit-Adjustment for Imbalanced Learning"
_NeurIPS.cc/2023/Conference — NeurIPS 2023 spotlight_

### Official Review · Reviewer_4AC3 · 2023-06-09

**Soundness:** 3 good
**Presentation:** 4 excellent
**Contribution:** 3 good
**Rating:** 7
**Confidence:** 4

**Summary:**

For class-imbalanced learning, many studies modified the loss function to emphasize the learning on minority classes by reweighting or logit adjustment. These studies showed high epirical classification performance, but existing generalization analysis of such studies is not unified. In this paper, the authors propose a data-dependent contraction to capture how the modified losses affect each class. With the data-dependent contraction technique, a fine-grained generalization bound is established and the authors analyze existing class imbalanced learning studies in a unified manner by applying the generalization bound to the VS loss. The authors propose two principled algorithm : TLA and ADRW based on the theoretical insights, and verify effectiveness of the two proposed algorithm through extensive experiments on benchmark datasets.

**Strengths:**

Well written paper, it is easy to follow the paper.

Great theoretical analysis and insights are given in this paper.

Novel algorithms are proposed based on the analysis and the insights.

Extensive experiments were conducted.

**Weaknesses:**

Code is not submitted. Thus I can not reproduce the experiment results in the paper.

Recently proposed class imbalanced learning algorithms based on multiple experts, contrastive learning, knowledge distillation are not reviewed, and also not compared in the section 4.3. It seems that the proposed algorithms achive lower classification performance than recently proposed algorithms such as PACO, RIDE and ACE.

**Questions:**

I have no questions.

**Limitations:**

The proposed algorithms (TLA, ADRW) improved classification performance, but the improvement seems not that significant under some settings (specifically between VS and VS+TLA+ADRW under CIFAR-10-LT, step w/sam and CIFAR-100-LT w/sam).

---

> ### Author Rebuttal · Authors · 2023-08-09
>
> > **Q1**: Code is not submitted. Thus I can not reproduce the experiment results in the paper.
>
> **A1**: Thanks for your constructive concern! According to the policy
>
> > If you were asked by the reviewers to provide code, please send an anonymized link to the AC in a separate comment (make sure the code itself and all related files and file names are also completely anonymized).
>
> we have sent the anonymized link to the AC. In this repository, we provide:
> - the **checkpoints and the log files** that exactly correspond to the results reported in the response **A2**, which is stored in the folder `existing_results`
> - the **code** that can reproduce the results, as well as the **bash script** stored in the file `bash_scripts/run.sh`. Since the proposed methods, i.e., ADRW and TLA work only when $t > T_0$, we resume from the checkpoint in the folder existing_results with $t = T_0$, such that the randomness can be eliminated. For example
>
> ```bash
> python cifar_train_sam.py --dataset cifar100 --imb_type exp --imb_factor 0.01 --loss_type VS --train_rule ADRW_T --gpu 1 --seed 0 --tro 0.25 --gamma 0.05 --tau 0.75 --rho 0.5 --wd 0.0005 --epochs 400 --t_reweight 320 --resume 'existing_results/better_wd+400epoch/320_ckpt.pth.tar' --exp_str Better_WD_400
> ```
>
> ---
>
> > **Q2**: Recently proposed class imbalanced learning algorithms based on multiple experts, contrastive learning, knowledge distillation are not reviewed, and also not compared in the section 4.3. It seems that the proposed algorithms achive lower classification performance than recently proposed algorithms such as PACO, RIDE and ACE.
>
> **A2**: Thanks very much for this constructive suggestion! Since this paper focuses on the loss-oriented methods for imbalanced learning, we mainly review prior works in this direction. According to your suggestion, we will update a more systematic review in the new version of the appendix to cover these orthogonal methods.
>
> We did not compare the mentioned methods since the experimental protocols are quite different. Taking CIFAR-100-LT ($\rho=100$) as an example, where `Simple Aug` denotes random crop and random flip, and `wd` represents weight decay.
>
> | | Ours | RIDE | ACE | PaCo |
> |:---|:---|:---|:---|:---|
> | Expert | Single | Multiple | Multiple | Single |
> | Training Epoch | 200 | 200 | 400 | 400 |
> | Data Augmentation | `Simple Aug` | `Simple Aug` + `Random Rotation` | `Simple Aug` + `Mixup` | `Simple Aug` + `RandAug` |
> | Better wd | $\times$ | $\checkmark$ | $\checkmark$ | $\checkmark$ |
>
> In fact, if we align these protocols, the proposed method can outperform these methods. Here are the results on CIAFR-100 LT ($\rho=100$), and we will provide more results in the future version.
>
> | Method | Balanced Accuracy |
> |:-|:-:|
> | RIDE | 49.10 |
> | ACE | 49.60 |
> | PaCo | 52.00 |
> | Ours | 46.40 |
> | Ours + Better wd | 49.54 |
> | Ours + Better wd + 400 epoch | 50.16 |
> | Ours + Better wd + 400 epoch + RandAugment | 52.97 |
>
> ---
>
> > **Q3**: The proposed algorithms (TLA, ADRW) improved classification performance, but the improvement seems not that significant under some settings (specifically between VS and VS+TLA+ADRW under CIFAR-10-LT, step w/sam and CIFAR-100-LT w/sam).
>
> **A3**: Thanks for your careful reading! Imbalanced learning is a challenging task, and even a small performance improvement is not trivial. In Sec.4, we reported the results averaged on 5 random seeds, and the proposed method outperforms the competitors consistently.
>
> Besides, we notice that the protocol might limit the improvement. After we adopt the protocol `Better wd + 400 epoch + RandAugment`, as mentioned in the response **A2**, for both VS and Ours, the improvement will be more significant:
>
> | Method | Balanced Accuracy |
> |:-|:-:|
> | VS | 51.83 |
> | Ours | 52.97 |

---

> > ### Comment · Reviewer_4AC3 · 2023-08-10
> > **I'll raise my score.**
> >
> > The authors clearly addressed my concern. Thus, I will raise my score as 7.

---

> > > ### Author Response · Authors · 2023-08-10
> > > **Thanks for the timely and kind comment**
> > >
> > > Thanks very much for your timely and kind comment! According to your suggestion, we will update more empirical results in the future version.

---

### Official Review · Reviewer_sr97 · 2023-07-03

**Soundness:** 4 excellent
**Presentation:** 3 good
**Contribution:** 4 excellent
**Rating:** 8
**Confidence:** 3

**Summary:**

In this paper, the authors study the problem of class imbalance. To be specific, they identify that there is a gap between the generalization theory of re-weighting & logit adjustment techniques and the practice. To be specific, they identify that the existing generalization bounds fail to account the imbalance among classes. The authors propose to close the gap by introducing an imbalance-specific bound, and then perform an analysis with existing methods to gain more insights about the effects of certain design choices. Finally, they introduce a variation on existing loss functions that perform better than existing approaches.

After the rebuttal:

With the rebuttal, the authors addressed my minor comments about clarifications on certain details. This is a solid NeurIPS paper. I recommend accepting the paper.

**Strengths:**

1. The paper provides theoretical as well as practical results & insights.

2. The four insights are very valuable.

3. Strong improvements over the baselines.

4. Generally easy to follow text.

**Weaknesses:**

I am generally happy with the paper. My only concern is that it contains many typos and grammatical errors. Moreover, I would find it easier if Fig 1, 2 and 3 analyzed the performances with respect to the imbalance ratio as commonly performed in the literature.


Minor comments:

- "[12] proposes an effective scheme" => Reference numbers should not be a part of a sentence. A correct way to write this is "Cao et al. [12] propose an effective scheme".

- The paper uses many many acronyms without introducing them.

- Lines 18-45: Already in these lines you should start with formulations (of ERM, margin theory etc.) since this paper is making theoretical contributions.

- Line 47: "loss function utilized in existing proof" => "loss function utilized in existing proofs"?

- Line 111: "Let ||.|| denotes" => "Let ||.|| denote".

- Line 114: "constants \mu" => "constant \mu".

- Line 207: "Do re-weigting and logit-adjustment fully compatible?" => "Are re-weigting and logit-adjustment fully compatible?"

**Questions:**

None.

**Limitations:**

Yes.

---

> ### Author Rebuttal · Authors · 2023-08-09
>
> Thanks very much for your nice suggestions, and we would like to make the following response.
>
> > **Q1**: Typos and grammatical errors.
>
> **A1**: Thanks for your careful reading! We will correct these typos and grammatical errors in the future version.
>
> ---
>
> > **Q2**: The paper uses many many acronyms without introducing them.
>
> **A2**: Thanks for this careful comment! We will check all the acronyms and provide their introduction in the future version:
> - The Class-Balanced (CB) loss [11]: This method proposes a re-weighting term that depends on the effective number of each class, where the effective number is based on the idea that diminishing marginal returns for additional samples.
> - The Label-Distribution-Aware Margin (LDAM) loss [12]: this loss function uses additive terms, which depend on the label distribution, to encourage the model to have the optimal trade-off between per-class margins.
> - The Deferred Re-Weighting (DRW) scheme [12]: This scheme deploys the re-weighting loss with a small learning rate after a vanilla ERM training period. In other words, the re-weighting phase is deferred.
> - The Logit-Adjustment (LA) loss [13]: this loss function uses additive terms to adjust the logits, such that the induced objective is Fisher consistent with the balanced accuracy.
> - The Class-Dependent Temperatures (CDT) loss [14]: this method introduces multiplicative terms, also named temperature, to compensate for the effect of feature deviation between training and test data.
> - The Vector-Scaling (VS) loss [16]: this loss function combines the advantages of both the additive terms of the LA loss and the multiplicative terms of the CDT loss.
>
> ---
>
> > **Q3**: Lines 18-45: Already in these lines you should start with formulations (of ERM, margin theory etc.) since this paper is making theoretical contributions.
>
> **A3**: Thanks for this constructive suggestion! In the future version, we will update the introduction to provide more formulations. For example, the naive Empirical Risk Minimization (ERM) can be denoted as
> $$
>     \min_f \hat{\mathcal{R}}(f) := \frac{1}{N} \sum_{(\boldsymbol{x}, y) \in \mathcal{S}} L(f(\boldsymbol{x}), y),
> $$
> where $L: \mathbb{R}^C \times \mathcal{Y} \to \mathbb{R}\_{+}$ measures the performance of the model $f: \mathcal{X} \to \mathbb{R}^C$ on the data point $(\boldsymbol{x}, y)$ belonging to the training set $\mathcal{S}$, and $C$ denotes the number of classes. As another example, we will provide the union bound based on the margin theory [12]:
> $$
>     \mathcal{R}\_\text{bal}(f) \precsim \frac{1}{C} \sum\_{y=1}^{C} \frac{1}{\text{margin}\_y^{\downarrow} \sqrt{N_y}},
> $$
> where $\text{margin}\_y^{\downarrow}$ represents the minimal margin of the class $y$, and $N_y$ denotes the number of samples in class $y$. In contrast, our bound can be formulated by:
> $$
>     \mathcal{R}\_\text{bal}(f) \precsim \frac{1}{C \pi_C} \sum\_{y=1}^{C} \mu_y \sqrt{\pi_y},
> $$
> where $\pi_y := N_y / N$, and $\mu_y$ is the local Lipschitz constant of $f$ for the class $y$. Next, we will point out that our result is shaper and can provide a series of insights, which can help highlight our theoretical contributions.
>
> ---
>
> > **Q4**: I would find it easier if Fig 1, 2 and 3 analyzed the performances with respect to the imbalance ratio as commonly performed in the literature.
>
> **A4**: Thank you very much for this constructive comment! We will update the results with respect to the imbalance ratio $\rho=10$ in the future version of Appendix. According to the policy, we first attach these figures in the PDF file of the “global” response.

---

> > ### Comment · Reviewer_sr97 · 2023-08-13
> > **Re: Rebuttal**
> >
> > Thank you for the rebuttal. I will keep my original recommendation as Strong Accept. Well done.

---

> > > ### Author Response · Authors · 2023-08-14
> > > **Thanks for this kind comment**
> > >
> > > Thanks very much for this kind comment! We will update our response to the future version, according to your suggestion.

---

### Official Review · Reviewer_D9Eg · 2023-07-04

**Soundness:** 3 good
**Presentation:** 3 good
**Contribution:** 3 good
**Rating:** 7
**Confidence:** 4

**Summary:**

This paper proposes a sharpened generalization bound of imbalance learning by directly bounding the balanced empirical risk. The authors achieve this by generalizing the Lipschitz Continuity to the Local Lipschitz Continuity with a group of constants, which, in VS Loss, is parameterized by a re-weighting term, a generalization term, and a logit adjustment term. By adjusting the above three terms with their proposed algorithm TLA and ADRW, the authors achieve keeping the balance between class balance and generalization of a model.

**Strengths:**

This paper unified re-weighting and logit adjustment, two common methods for solving the imbalance problem, which I think is a good contribution to this field. Moreover, the authors' entry point is novel, providing sophisticated proof for their theory, the paper is well-written, and the experiments also well demonstrate their claims.

**Weaknesses:**

The authors only perform their experiment on the ResNet family for all the baselines and their proposed method. I hope the authors can provide the experiment result of their proposed method on different network structures in future work.

**Questions:**

I have no questions from the authors.

**Limitations:**

The authors have discussed their limitations in the form of future work in Section 5.

---

> ### Author Rebuttal · Authors · 2023-08-09
>
> Thanks very much for your nice suggestions, and we would like to make the following response.
>
> > **Q**: The authors only perform their experiment on the ResNet family for all the baselines and their proposed method. I hope the authors can provide the experiment result of their proposed method on different network structures in future work.
>
> **A**: Thank you for this nice suggestion! As mentioned in Appendix E, we follow the protocols of the prior arts [12, 28, 37]. To further investigate the effect of backbone, we conduct a preliminary experiment based on DenseNet121 on CIFAR-100 LT ($\rho=100$). Due to the time limit, we only tune $\alpha_y, \beta_y, \Delta_y$ and fix the other parameters such as the learning rate, weight decay, and the training epoch. The empirical results averaged on three seeds are listed as follows, which are consistent with those of ResNet. We will provide more results in future work.
>
> | Dataset | CIFAR-100 |
> |:------- |:---------:|
> | Imbalance Type | LT ($\rho=100$) |
> | CE | 42.8 $\pm$ 0.3 |
> | CE + DRW | 44.1 $\pm$ 0.6 |
> | CE + ADRW | 45.2 $\pm$ 0.4 |
> | VS | 47.5 $\pm$ 0.5 |
> | VS + DRW | 47.5 $\pm$ 0.3 |
> | VS + TLA + ADRW | 48.0 $\pm$ 0.4 |

---

### Official Review · Reviewer_2B2B · 2023-07-05

**Soundness:** 3 good
**Presentation:** 3 good
**Contribution:** 3 good
**Rating:** 6
**Confidence:** 4

**Summary:**

This paper provides a unified generalization analysis of the loss-modification approaches for imbalanced learning. It analyzes the gap between balanced accuracy and empirical loss (the loss may involve re-weighting and logit-adjustment approaches). It further provides empirical analysis that matches the theoretical insights. And a new method induced by the theoretical results with better performance is also proposed.

**Strengths:**

1. This paper directly establishes a theoretical connection between balanced accuracy and empirical loss, which is not attained in previous work.
2. In this paper, the existing methods are systematically reviewed, and according to the new theoretical results, the existing methods are well classified and discussed.
3. The charts and tables are beautifully formatted and laid out.


**Weaknesses:**

1. What we would like to bound is the balanced accuracy $\mathcal{R}_{bal}$, where the loss is measured by $M$. However, in the imbalanced learning setting, $L$ is not just a differential surrogate version of $M$, but instead is an adjusted version that puts more emphasis on the small classes. In such circumstances, what is the meaning of bounding $\mathcal{R}_{bal}^L$? Measuring under a class-balanced distribution while small classes are still emphasized in the loss? (By the way, please check the mapping of L in line 84.)
2. How is ‘data-dependent’ exhibited in the theoretical result? Is it just about the sample size in each class, i.e., label distribution? Is there anything more than label distribution?
3. In Eq.(14), what is By(f)? I checked the supplementary and I see it is a constant. What is the meaning of it? Is it the minimal margin on class y? I think these points should be made clear in the main body of the paper. And at the moment, I am also doubtful about the argument ‘balanced / imbalanced By(f)’ in the paragraph of line 192-200, which may involve the mechanisms in By(f).
4. Maybe I misunderstood it, I suppose in Figure 3, T0 marks the start of DRW, and the total epoch T is fixed. Then why the line ‘CE+None’ is not constant in Figure 3(b) and 3(c)？
5. In section 3.3, there is no mention of the setting of $\Delta_y$. So does the setting of $\beta_y$ in the $t<T_0$ stage. And in experiments, it would be better if provided more details on how these hyperparameters are tuned.


**Questions:**

SEE Weaknesses.

**Limitations:**

NAN.

---

> ### Author Rebuttal · Authors · 2023-08-09
>
> Thanks for your constructive comments, and the response is as follows.
>
> > **Q1**: The meaning of bounding $\mathcal{R}_{bal}^L$.
>
> **A1**: In the `non-asymptotic` level (finite samples), bounding $\mathcal{R}\_{bal}^L$ can put more emphasis on minority classes, which is consistent with your understanding. But if we take an `asymptotic` view (infinite samples), it is beyond an adjusted version. Ideally, a small $\mathcal{R}\_{bal}^L$ should induce a small $\mathcal{R}\_{bal}$. To this end, a basic requirement is Fisher consistency. That is, optimizing $\mathcal{R}\_{bal}^L$ can recover the optimal solution to $\mathcal{R}\_{bal}$:
> $$
>     \mathcal{R}\_{bal}^L(f) \to \min\_f \mathcal{R}\_{bal}^L(f) \Rightarrow  \mathcal{R}\_{bal}(f) \to \min\_f \mathcal{R}\_{bal}(f).
> $$
> As mentioned in Sec.2.1, the generalized loss we consider has a consistent special case [13]:
> $$
>     L\_F(\boldsymbol{s}, y) := \frac{\delta\_y}{\pi\_y} \log [1 + \sum\_{y' \neq y} \frac{\delta\_{y'}}{\delta\_{y}} e^{\boldsymbol{s}\_{y'} - \boldsymbol{s}\_{y}}],
> $$
> where $\delta_y > 0$ is an arbitrary constant. Hence, if we select such a $L$, bounding $\mathcal{R}\_{bal}^L$ also helps bound $\mathcal{R}_{bal}$. Of course, not all losses are consistent, and we need a systematic analysis in future work.
>
> ---
>
> > **Q2**: Check the mapping of $L$.
>
> **A2**: Thanks for your careful reading! The mapping of $L$ should be $\mathbb{R}^C \times \mathcal{Y} \to \mathbb{R}_+$. We will correct it in the future version.
>
> ---
>
> > **Q3**: How is data-dependent exhibited?
>
> **A3**: Compared with the union bound [12], data-dependent is exhibited in:
> - The basic lemma, *i.e.*, Lemma 2 introduces $1 / \pi_C$ to the generalization bound. Since $\pi_C$ denotes the ratio of the most minor class, it can be regarded as a measure of imbalance degree. In other words, this lemma shows how the model performance depends on the imbalance degree of the data.
> - Based on the proposed techniques, Theorem 1 and Proposition 3 reveal how existing loss-oriented methods improve generalization performance by exploiting data priors, which is also beyond the label distribution itself.
>
> ---
>
> > **Q4**: The meaning of $B_y(f)$ and 'balanced/imbalanced $B_y(f)$'.
>
> **A4**: Perhaps due to the way of our writing, it is a pity to leave the impression that $B_y(f)$ is not well formulated. As described in line 462, $B_y(f)$ is the minimal prediction on the ground-truth class $y$, i.e., $B_y(f) := \min\_{\mathbf{x} \in S_y} s_y$. Following your understanding, $B_y(f)$ is closely related to the minimal margin, which is defined as $\text{margin}\_y^\downarrow := \min_{\mathbf{x} \in S_y} (s_y - \max_{j \neq y} s_j)$. In other words, **a large $B_y(f)$ can implicitly increase the minimal margin of the class $y$**. Meanwhile, we have
> $$
>     B_y(f) - \text{margin}\_y^\downarrow \le \max\_{\mathbf{x} \in S_y, j \neq y} s_j.
> $$
> Hence, **as we improve the model performance on class $y$, the RHS of the above inequality, i.e., the gap between $B_y(f)$ and $\text{margin}_y^\downarrow$ will decrease, and both the minimal margin and $B_y(f)$ will increase.**
>
> Keeping this mechanism in mind, the argument 'balanced/imbalanced $B_y(f)$' becomes easier to understand. In fact, `balanced/imbalanced` are indeed a little confusing, thus we next use `take into account`. As shown in Fig.3a, a weighted loss can boost the model performance on minority classes but hinders further improvement on majority classes. As a result, majority/minority classes have relatively small/large $B_y(f)$, respectively (*i.e.*, fail to take into account the $B_y(f)$ of majority classes). By contrast, DRW helps both majority classes and minority classes have a small $B_y(f)$ (*i.e.*, take into account $B_y(f)$ of both majority classes and minority classes). This explains why DRW can bring a better generalization performance, which is our main point.
>
> ---
>
> > **Q5**: The line 'CE+None' is not constant in Figure 3.
>
> **A5**: In DRW, both reweighting and small learning rates will be used when $t > T_0$ [12]. To provide a more informative comparison, we also decrease the learning rates of `CE+None` at the corresponding epoch $T_0$ as `CE+DRW`, making the line in Fig.3 not constant. We will clarify this issue in the future version.
>
> ---
>
> > **Q6**: The setting of $\Delta_y$ and $\beta_y$.
>
> **A6**: Perhaps due to the way of our writing, it is a pity to leave the impression that $\beta_y$ and $\Delta_y$ are not well defined in Sec.3.3. In fact, our analysis and the induced algorithm are both based on the generalized loss defined in Sec.2.1. Hence, all the $\beta_y$ and $\Delta_y$ mentioned in Sec.2.1 are reasonable options. In general, the loss family can be written as:
> $$
>     L_\text{VS}(\boldsymbol{s}, y) = - \alpha_y \log \left( \frac{e^{\beta_y s_y + \Delta_y}}{\sum_{y'} e^{\beta_{y'} s_{y'} + \Delta_{y'}} } \right).
> $$
> And
> - $\beta_y \in \{1, (N_y / N_1)^{\gamma}\}$, where $N_y$ denotes the number of samples in the class $y$, and $\gamma > 0$.
> - $\Delta_y \in \{0, \tau \log \pi_y, - \frac{C}{N_y^{1 / 4}} \}$, where $\tau, C >0$. If $\Delta_y = - \frac{C}{N_y^{1 / 4}}$ for the ground-truth label $y$, $\Delta_{y'} = 0$ for $y' \neq y$ [12].
>
> We will clarify this issue in the future version.
>
> ---
>
> > **Q7**: Details of hyperparameter search.
>
> **A7**: To validate our theoretical results, we first tune these parameters via grid search as suggested in [13, 16]. Specifically, $\alpha_y \propto \pi_y^{-\nu}$, and $\nu$ is searched in $\\{0.15, 0.25, 0.75, 1.0, 2.0, 3.0\\}$; $\Delta_y = \tau \log \pi_y$, and $\tau$ is searched in $\\{0.5, 0.75, 1.0, 1.25, 2.0\\}$; $\beta_y = (N_y / N_1)^\gamma$, and $\gamma$ is searched in $\\{0.05, 0.1, 0.15, 0.2, 0.25\\}$.
>
> Benefiting from the theoretical validation in Sec.4.2, the complexity can be significantly decreased. For example, according to Fig.4, we will choose a small $\gamma$ when $\nu$ is large to avoid the incompatibility issue. We will update this detail in the future version.

---

> > ### Comment · Reviewer_2B2B · 2023-08-20
> >
> > Thank you for your answer and your clarifications. I will keep my score unchanged.

---

> > > ### Author Response · Authors · 2023-08-20
> > > **Thanks for your kind comment**
> > >
> > > Thanks for your kind comment! According to your suggestion, we will update our response to the future version to clarify these issues.

---

### Author Rebuttal · Authors · 2023-08-09

Dear reviewers,

First, we would like to express our sincere gratitude for your valuable comments. Following the valuable suggestions, we have carefully polished and improved the corresponding details. Now we present a brief summary of the response.

- **We clarify some important concepts** such as
    - the meaning of bounding $\mathcal{R}_\text{bal}^L$, (**Q1** for Reviewer 2B2B)
    - the meaning of 'data-dependent', (**Q3** for Reviewer 2B2B)
    - the mechanisms of $B_y(f)$, (**Q4** for Reviewer 2B2B)
    - the line of 'CE+None' in Fig.3, (**Q5** for Reviewer 2B2B)
    - the setting of $\beta_y, \Delta_y$, (**Q6** for Reviewer 2B2B)
    - the strategy for tuning the hyper-parameters, (**Q7** for Reviewer 2B2B)
    - the meaning of acronyms. (**Q2** for Reviewer sr97)
    - the formulation provided in the future version of the introduction. (**Q3** for Reviewer sr97)
- **We provide more empirical results**, including
    - those with different backbones (DenseNet121), (**Q** for Reviewer D9Eg)
    - those with a imbalance ratio $\rho=10$ (Fig.1-3), (**Q4** for Reviewer sr97)
    - those with different protocols (weight decay, epochs, and data augmentation). (**Q2,Q3** for Reviewer 4AC3)
- **We uploaded the code to an anonymized repository and sent its link to AC**. (**Q1** for Reviewer 4AC3)
- **We checked the typos and grammatical errors**. (**Q2** for Reviewer 2B2B, **Q1** for Reviewer sr97)

Please refer to the respective response for more details, and we will update all these improvements in the future version.

---

### Decision · Program_Chairs · 2023-09-21

**Decision:**

Accept (spotlight)

**Comment:**

Reviewers were unanimously supportive of this paper, which provides an interesting new analysis of re-weighting and logit adjustment for learning with label imbalance. The paper makes solid theoretical and empirical contributions, and is likely to be of broad interest to the community.